# Long-Term Monthly 0.05° Terrestrial Evapotranspiration Dataset (1982–2018) for the Tibetan Plateau

Ling Yuan [1,2], Xuelong Chen[1,4,6*], Yaoming Ma[1,2,3,4,5,6*], Cunbo Han[1,4,6], Binbin Wang[1,4,5,6], Weiqiang Ma[1,4,6]

[1] State Key Laboratory of Tibetan Plateau Earth System, Environment and Resources (TPESER), Institute of Tibetan Plateau Research, Chinese Academy of Sciences, Beijing 100101, China.

[2] College of Earth and Planetary Sciences, University of Chinese Academy of Sciences, Beijing 100049, China

[3] College of Atmospheric Science, Lanzhou University, Lanzhou 730000, China

[4] National Observation and Research Station for Qomolongma Special Atmospheric Processes and Environmental Changes, Dingri 858200, China

[5] Kathmandu Center of Research and Education, Chinese Academy of Sciences, Beijing 100101, China

[6] China-Pakistan Joint Research Center on Earth Sciences, Chinese Academy of Sciences, Islamabad 45320, Pakistan

**Corresponding author and address:**

Xuelong Chen, Dr., Prof., x.chen@itpcas.ac.cn

Yaoming Ma, Dr., Prof., ymma@itpcas.ac.cn

Building 3, No.16 Lincui Road, Chaoyang District, Beijing 100101, China

**Abstract**

Evapotranspiration (ET) plays a crucial role in the water balance of the Tibetan Plateau (TP), often referred to as the "Asian water tower" region. However, accurately monitoring and comprehending the spatial and temporal variations of ET components (including soil evaporation $E_s$, canopy transpiration $E_c$, and intercepted water evaporation $E_w$) in this remote area remains a significant challenge due to the limited availability of observational data. This study generates a 37-year dataset (1982–2018) of monthly ET components for the TP using the MOD16-STM (MOD16 soil texture model). This model utilizes up-to-date soil properties, meteorological data, and remote sensing datasets. The estimated ET results strongly correlate with measurements from nine flux towers, demonstrating a low root mean square error of 13.48 mm/month, a mean bias of 2.85 mm/month, a coefficient of determination of 0.83, and an index of agreement of 0.92. The annual average ET for the entire TP, defined as elevations higher than 2500 meters, is approximately 0.93±0.037 Gt/year. The predominant contributor to ET on the TP is $E_s$, accounting for 84% of the total ET. Our findings reveal a noteworthy upward trend in ET in most central and eastern parts of the TP, with a rate of approximately 1–4 mm/year ($p<0.05$) and a significant downward trend with rates between −3 and 1 mm/year in the northwestern part of TP during the period from 1982 to 2018. The average annual increase in ET for the entire TP over the past 37 years is approximately 0.96 mm/year. This upward trend can be attributed to the TP's warming and wetting climate conditions. The MOD16-STM ET dataset demonstrates a reliable performance across the TP compared to previous research outcomes. This dataset is valuable for research on water resource management, drought monitoring, and ecological studies. The entire dataset is freely accessible through the Science Data Bank (http://doi.org/10.11922/sciencedb.00020, Y. Ma*, X. Chen*, L. Yuan, 2021) and the National Tibetan Plateau Data Center (TPDC) (https://data.tpdc.ac.cn/en/disallow/e253621a-6334-4ad1-b2b9-e1ce2aa9688f/, http://doi.org/10.11888/Terre.tpdc.271913, L. Yuan, X. Chen*, Y. Ma*, 2021).

**Keywords:** Evapotranspiration; MOD16-STM; Climate change; Asian water tower; Tibetan Plateau

## 1. Introduction

The Tibetan Plateau (TP) (24–40°N, 70–105°E) is often referred to as the "Asian water tower" owing to its distinctive geographical and ecological characteristics, as acknowledged in studies by Immerzeel et al. (2010, 2020), Yao et al. (2012), and Xu et al. (2019). Within this region, evapotranspiration (ET) plays a vital role in the overall water balance. The TP predominantly features grassland (covering more than 47% of the area) and sparse vegetation or bare soil (surrounding over 33%), as indicated by the Moderate Resolution Imaging Spectroradiometer (MODIS) land cover dataset (MCD12C1) (Fig. 1c). Arid or semi-arid conditions mostly characterize this vast expanse. The TP is currently undergoing significant changes in its hydrological cycle, driven by global warming, as documented in studies by Yang et al. (2014), Kuang et al. (2016), and Zohaib et al. (2017). Nevertheless, accurately monitoring the spatial and temporal fluctuations in ET on the TP remains a formidable challenge due to the intricate environmental conditions of the TP. Moreover, understanding how ET patterns on the TP will evolve in the context of global warming is essential for assessing the impacts of these changes on the local population's livelihoods.

In recent years, various datasets for estimating ET on the TP have been developed, including the complementary relationship (CR) model (Ma et al., 2019; Wang et al., 2020), the surface energy balance system (SEBS) model (Chen et al., 2014, 2021; Zhong et al., 2019; Han et al., 2017, 2021), and the Penman–Monteith model with remote sensing (RS-PM) (Wang et al., 2018; Song et al., 2017; Chang et al., 2019; Ma et al., 2022), among others. However, a considerable variance exists among these TP ET products (Peng et al., 2016; Baik et al., 2018; Li et al., 2018; Khan et al., 2018). Studies have utilized eddy-covariance measurements (Shi et al., 2014; You et al., 2017; Yang et al., 2019; Ma et al., 2020) and reanalysis datasets (Shi et al., 2014; Dan et al., 2017; Yang et al., 2019; De Kok et al., 2020) to investigate ET on the TP. A recent Han et al. (2021) study produced the region's 18-year ET dataset (2001–2018). Enhancements to the canopy conduction algorithm in the Penman–Monteith model have led to improved ET estimates in previous research (Leuning et al., 2008; Zhang et al., 2010; Li et al., 2015; Zhang et al., 2016, 2019; Gan et al., 2018). However, these ET products tend to perform poorly in TP areas with sparse vegetation and arid to semi-arid climates (Zhang et al., 2010; Li et al., 2014b; Song et al., 2017; Baik et al., 2018; Li et al., 2018; Khan et al., 2018).

The limitations of the MOD16 Penman–Monteith model in arid to semi-arid TP regions are primarily due to its failure to consider the dominant role of topsoil texture and topsoil moisture in governing $E_s$ processes (Yuan et al., 2021). Accurately separating and validating ET components on the TP remains challenging, even

though total ET estimates tend to align across different products (Lawrence et al., 2007; Blyth and Harding,
2011; Miralles et al., 2016). The TP is primarily characterized by short and sparse vegetation, and soil moisture
is crucial in ET estimation for this region. Several studies have used the Penman–Monteith algorithm to estimate
ET on the TP (Wang et al., 2018; Ma et al., 2022). However, these studies have not accounted for the effects of
soil moisture ($SM$) on evaporation resistance and stomatal conductance.
The enhanced Penman–Monteith model, MOD16-STM (MOD16 soil texture model), has been developed
to address these limitations. MOD16-STM redefines the modules for $E_s$ to consider the impacts of $SM$ on soil
evaporation resistance. This modification is based on eddy-covariance (EC) observations conducted on the TP
(Yuan et al., 2021), offering a promising opportunity to estimate ET components in this region accurately. $E_s$
often dominates ET in sparsely vegetated areas, especially in arid and semi-arid regions with large bare soil
areas (Wilcox et al., 2003; Kool et al., 2014; Wang et al., 2018; Ma et al., 2015; Ma and Zhang, 2022). Previous
studies have highlighted that 20% to 40% of global ET is attributed to $E_s$. Bare soil surface evaporation is a
rapid process influenced by shallow surface water (Koster and Suarez, 1996). $E_s$ is primarily controlled by water
diffusion in the soil (Good et al., 2015; Yuan et al., 2022). Accurately quantifying and separating $E_s$ is crucial
for enhancing our understanding of water and energy cycles on the TP. However, quantifying ET and its
components remains challenging due to the influence of atmospheric demand, soil moisture conditions, and
complex interactions between heterogeneous vegetation and soil properties (Merlin et al., 2016; Wu et al., 2017;
Philips et al., 2017; Lehmann et al., 2018). MOD16-STM holds the potential to generate a remote sensing $E_s$
and ET component dataset covering the satellite era since 1980. In this study, the MOD16-STM model,
acknowledging its limitations, was employed to estimate a long-term ET and ET components dataset spanning
37 years (1982–2018) (Yuan et al., 2021).
A preferable approach involves directly estimating ET based on topsoil moisture, significantly impacting
the TP's surface water exchange. Thus, leveraging the advantages of the MOD16-STM model for ET estimation
on the TP, this study aimed to achieve two main objectives: (1) develop a 37-year (1982–2018) monthly ET
dataset for the TP at a 0.05°×0.05° spatial resolution; (2) quantify the spatial distribution and spatiotemporal
variability of ET and its components across the TP.
**2. Materials and methods**
**2.1 Study area**
The Tibetan Plateau, located between 25–40°N and 74–104°E, spans approximately 2.5 million km$^2$ and

consists of land above 2,500 m in altitude (Fig. 1a). This region, as indicated by the FAO drought index dataset, represents the largest landform unit in Eurasia and encompasses hyper-arid, arid, semi-arid, and sub-humid climate zones (Fig. 1b). The land cover types primarily include mixed forests, grasslands, bare soil, glaciers, and snow-covered areas (see Fig. 1c). The topsoil predominantly consists of sandy loam, loam, and clay (Fig. 1d). The annual average temperature in the region ranges from approximately −3.1°C to 4.4°C. Average annual precipitation gradually increases from less than 50 mm in the northwest to over 1000 mm in the southeast, with the most precipitation occurring during summer (Ding et al., 2017). Over time, the TP has undergone significant environmental changes, including increased precipitation, decreased wind speed (*wind*), fewer snow days, reduced radiation, thawing permafrost, glacier melting, and increased vegetation (Kang et al., 2010; Yao et al., 2012; Yang et al., 2014; Kuang et al., 2016; Bibi et al., 2018; Chen et al., 2019).

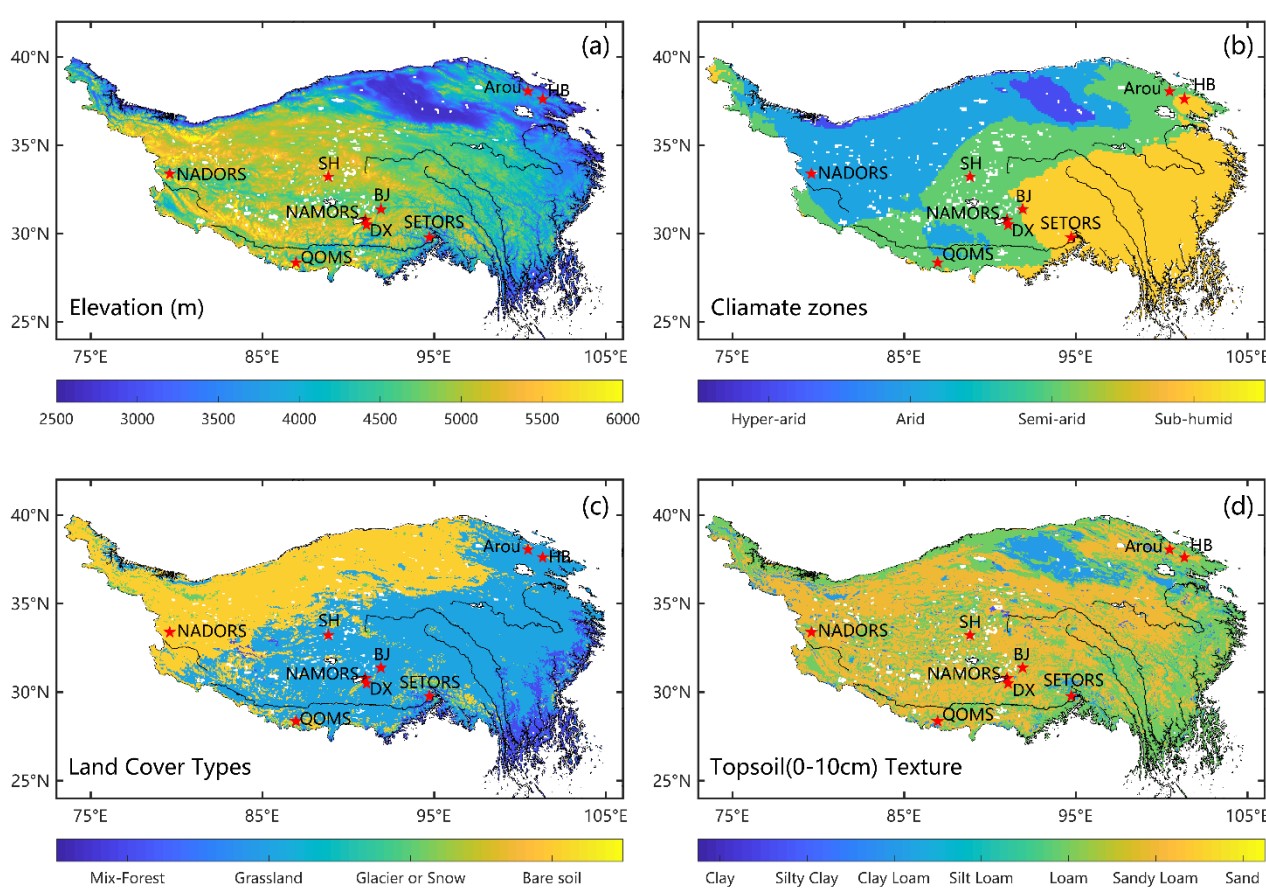

Figure 1. Maps of the (a) topography (STRM), (b) climate zones (FAO aridity index), (c) land cover types (MCD12C1), and (d) soil textures (HWSD) in the study area. The red dots indicate the flux site locations.

**2.2 Generation of a long-term series of monthly ET products**

This study introduces a novel dataset comprising a long-term series of monthly ET generated using the

MOD16-STM model. The process of calculating monthly ET with the MOD16-STM model and the associated
driving datasets is illustrated in Figure 2.

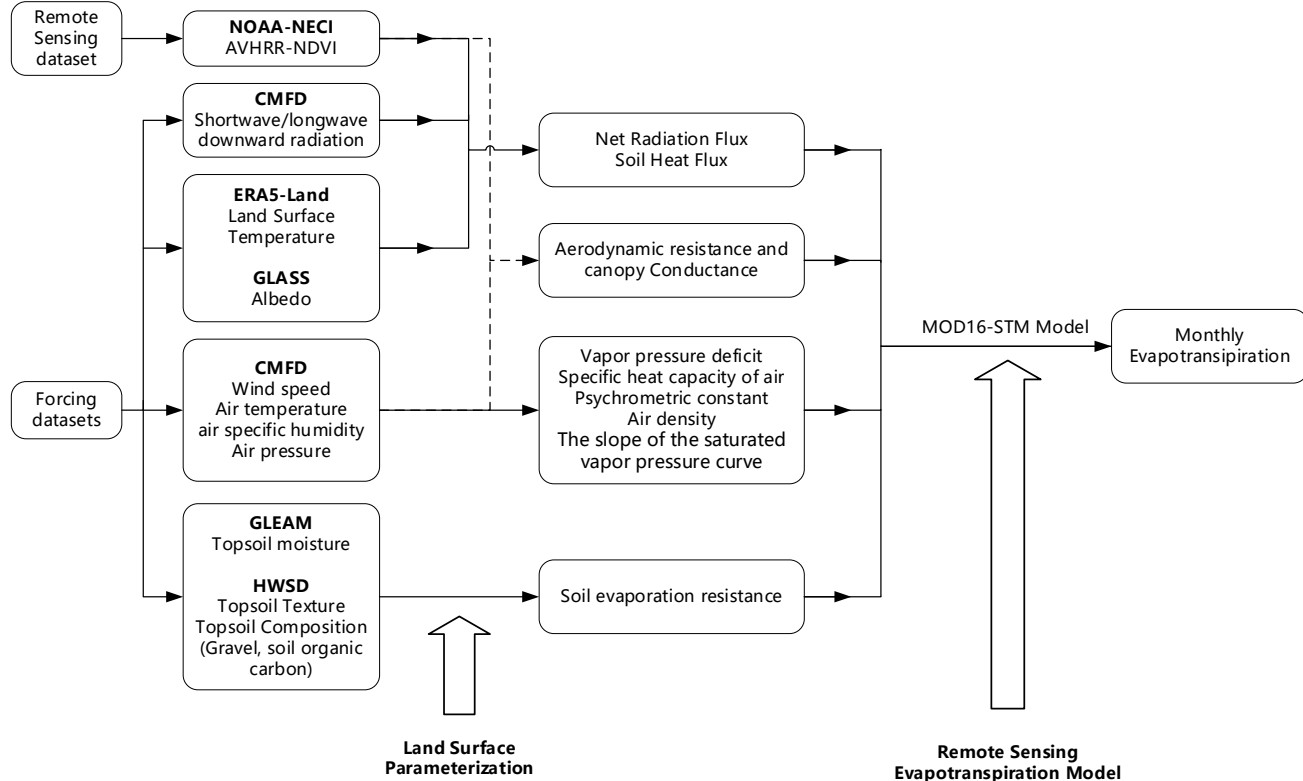


Figure 2. Workflow of the MOD16-STM evapotranspiration product.

**2.2.1 Description of MOD16-STM ET model**

The MOD16-STM model computes the ET components using the Penman–Monteith equation as follows:

$$E_c = \frac{(\Delta \times f_c \times (R_n - G_0) + \rho_a \times C_p \times \frac{VPD}{r_a} \times f_c) \times (1 - F_{wet})}{\lambda \times \left(\Delta + \gamma \times (1 + \frac{r_c}{r_a})\right)} \tag{1}$$

$$E_s = \frac{(\Delta \times (1 - f_c) \times (R_n - G_0) + \rho_a \times C_p \times \frac{VPD}{r_a}) \times (1 - F_{wet})}{\lambda \times \left(\Delta + \gamma \times (1 + \frac{r_s}{r_a})\right)} \times \left(\frac{RH}{100}\right)^{\frac{VPD}{\beta}} \tag{2}$$

$$E_w = E_{wet\_s} + E_{wet\_c} \tag{3}$$

The total ET combines three distinct components: $E_c$, $E_s$, and $E_w$ (wet surface evaporation). For a more detailed
explanation of the calculations for $E_{wet\_s}$ (evaporation from wet soil) and $E_{wet\_c}$ (evaporation from wet canopy),
you can refer to Yuan et al. 2021. Here, $r_a$ (s/m) is the aerodynamic resistance, $r_c$ (s/m) is the aerodynamic
resistance of water vapor of the canopy, and $r_s$ (s/m) is the surface (or canopy) resistance. Yuan et al. (2021)
optimized MOD16 $r_a$ based on the Monin-Obukhov similarity theory (MOST) and calibrated the empirical
values of $r_c$ for grassland underlying surfaces. They also pointed out that the topsoil moisture content directly
affects the value of $r_s$, indirectly influencing the $E_s$ process. Therefore, this study extended this optimization
algorithm from the site scale to the regional scale. The variables used in the above equations are defined as
follows:

•    $R_n$ represents the net radiation flux (W/m²).

•    $G_0$ denotes the soil heat flux (W/m²).

•    $\rho_a$ is the density of the air (kg/m³).

•    $C_p$ stands for the specific heat capacity of the air (J/(kg·K)).

•    $VPD$ represents the vapor pressure deficit (hPa).

•    $\Delta$ represents the slope of the saturated vapor pressure curve (hPa/K).

•    $\gamma$ is the psychrometric constant (hPa/K), calculated as $\gamma = C_p \cdot P_a \cdot M_a / (\lambda \cdot M_w)$, where $\lambda$ is the latent heat

of vaporization (J/kg), and $M_a$ and $M_w$ are the molecular masses of dry air and wet air, respectively.

•    $r_a$ signifies the aerodynamic resistance (s/m).

•    $r_s$ represents the surface (or canopy) resistance (s/m).

•    $F_{wet}$ is the relative surface wetness.

•    The vegetation cover fraction ($f_c$) is estimated using the NDVI (Normalized Difference Vegetation

Index).

$$f_c = \left( \frac{NDVI - NDVI_{\min}}{NDVI_{\max} + NDVI_{\min}} \right)^2 \tag{4}$$

$R_n$ and $G_0$ are calculated as follows:

$$R_n = (1 - \alpha) \times SWD + LWD - \varepsilon \times \sigma \times LST^4 \tag{5}$$

$$G_0 = R_n \times (I_c + (1 - f_c) \times (I_s - I_c)) \tag{6}$$

Here, $SWD$ is the downward shortwave radiation, $\alpha$ is land surface albedo, $LWD$ is the downward longwave
radiation, $\sigma$ represents the Stefan-Boltzmann constant ($5.67 \times 10^{-8}$ W/(m²·K⁴)), $\varepsilon$ is emissivity, and $LST$ means
land surface temperature. $I_c$ (= 0.05) and $I_s$ (= 0.315) are the ratios of ground heat flux and net radiation for
surfaces with full vegetation cover (Su et al., 2002) and bare soil (determined by NDVI<0.25 in this study)
(Yuan et al., 2021), respectively. When the air temperature ($T_a$) is below 5°C, photosynthesis and transpiration
processes are not active, and therefore, $E_c$ is not considered in the calculations. When the land surface
temperature is below 0°C, the sublimation equation is derived by modifying the surface energy balance equation
using the Clausius–Clapeyron equation, accounting for the equilibrium of water vapor in both liquid and frozen
states. It's important to note that this study did not estimate the evaporation from water surfaces. Previous
research has extensively examined water surface evaporation from lakes on the Tibetan Plateau in detail (Wang
et al., 2020). Therefore, this study focuses on land ET estimation, excluding water surface evaporation.
Numerous prior studies have employed optimized conductance to estimate $E_c$ (Jarvis et al., 1976; Irmak
and Mutiibwa, 2010; Zhang et al., 2010; Leuning et al., 2008; Li et al., 2013, 2015), as well as $E_s$ (Sun et al.,
1982; Camillo and Gurney, 1986; Sellers et al., 1996; Sakaguchi and Zeng, 2009; Ortega-Farias et al., 2010;
Tang et al., 2013). This study computed the $r_a$ using the MOST (Thom, 1975; Liu et al., 2007).

$$r_a = \frac{\ln\left(\frac{z_h - d_0}{z_{0h}} - \Psi_h\right)\ln\left(\frac{z_m - d_0}{z_{0m}} - \Psi_m\right)}{k^2 u} \tag{7}$$

Where $k$ represents the von Karman's constant (0.41), $z_h$ and $z_m$ denote the measurement heights for $T_a$ and $wind$,
and $d_0$ represents the displacement height. The stability correction functions for momentum ($\psi_m$) and heat
transfer ($\psi_h$) can be computed using universal parts. These correction terms' mathematical expressions (Eq. 8–
12) are as follows (Högström, 1996; Paulson, 1970).
For stable conditions:

$$\psi_m = -5.3\frac{(z_m - z_{0m})}{L} \tag{8}$$

$$\psi_h = -8.0\frac{z_h - z_{0h}}{L} \tag{9}$$

For unstable conditions:

$$\psi_m = 2\ln\left(\frac{1+x}{1+x_o}\right) + \ln\left(\frac{1+x^2}{1+x_o^2}\right) - 2\tan^{-1}x + 2\tan^{-1}x_o \tag{10}$$

$$\psi_h = 2\ln\left(\frac{1+y}{1+y_o}\right) \tag{11}$$

For neutral conditions:

$$\psi_m = \psi_h = 0 \tag{12}$$

In Equations (8–11), the following variables and parameters are defined: $x = (1-z_m/L)^{0.25}$, $x_o = (1-z_{om}/L)^{0.25}$, $y =$
$(1-11.6\ z_h/L)^{0.5}$, and $y_o = (1-11.6\ z_{oh}/L)^{0.5}$ (Högström, 1996; Paulson, 1970). Here, $L$ represents the Obukhov
length (m), calculated as $L = T_a \cdot u_*^2/(k \cdot g \cdot T_*)$, where $g = 9.8$ m/s² and $T_*$ is the fractional temperature (K) and
$u_*$ denotes the friction velocity (m/s). $T_*$ is further defined as $T_*=-(\theta_s-\theta_a)/((\ln(z_h/z_{oh})-\psi_h)$, where $\theta_s$ can be
approximated using the *LST,* and $\theta_a=T_a+z_h \cdot g/C_p$. The parameterization of $u_*$ and $L$ has been successfully applied
in previous studies on the TP (Chen et al., 2013). $z_{0h}$ represents the roughness length for heat transfer (m). A
parameterization scheme for $z_{0h}$ developed by Yang et al. (2008) has been widely utilized in remote sensing land
surface fluxes and land surface models (LSMs) across the TP (Biermann et al., 2014; Chen et al., 2013; Ma et
al., 2015). This scheme has also been employed in the current study for consistency.

$$z_{0h} = \frac{70v}{u_*} exp(-7.2u_*^{0.5}|T_*|^{0.25}) \tag{13}$$

where $v$ is the fluid kinematic viscosity, $v=1.328\times10^{-5} \cdot (P_0/P_a) \cdot (T_a/T_0)^{1.754}$, $P_0=1013$ hPa and $T_0=273.15$ K. The
roughness height for momentum transfer ($z_{0m}$) was determined based on canopy height ($h_c$), following the
method outlined by Chen et al. (2013). The water saturation degree of surface soil ($SM/\theta_{sat}$) is utilized to impose
soil classification and soil texture constraints on the $r_s$ and $E_s$ estimates (Yuan et al., 2021), as follows:

$$r_s = exp\left(a + b \times \frac{SM}{\theta_{sat}}\right) \tag{14}$$

Here, the parameters $a$ and $b$ are empirical coefficients that vary based on different soil textures, as documented
in Table 1. The estimation of $\theta_{sat}$, which considers soil organic content (SOC) and gravel content, can be
obtained using the Soc-Vg scheme (Chen et al., 2012; Zhao et al., 2018), and its calculation is as follows:

$$\theta_{sat} = (1-V_{SOC}-V_g)\times\theta_{sat,m}+V_{SOC}\times\theta_{sat,sc} \tag{15}$$

Where $\theta_{sat,m}$ represents the porosity of the mineral soil and can be calculated as $\theta_{sat,m} = 0.489-0.00126 \cdot \%sand$
(Cosby et al., 1984). Additionally, $\theta_{sat,sc}$ is the porosity of the SOC. It is assumed to be 0.9 m³·m⁻³ in this study,
as per the work of Farouki (1981) and Letts et al. (2000). The variables $V_{soc}$ and $V_g$ denote the volumetric
fractions of the SOC and gravel, respectively, and their calculation is as follows:

$$V_{SOC} = \frac{\rho_p \times (1-\theta_{sat,m}) \times m_{SOC}}{\rho_{SOC} \times (1-m_{SOC}) + \rho_p \times (1-\theta_{sat,m}) \times m_{SOC} + (1-\theta_{sat,m}) \times \frac{\rho_{SOC} \times m_g}{1-m_g}} \tag{16}$$

$$V_g = \frac{\rho_{SOC} \times (1-\theta_{sat,m}) \times m_g}{(1-m_g) \times \left(\rho_{SOC} \times (1-m_{SOC}) + \rho_p \times (1-\theta_{sat,m}) \times m_{SOC} + (1-\theta_{sat,m}) \times \frac{\rho_{SOC} \times m_g}{1-m_g}\right)} \tag{17}$$

In these equations, $\rho_p$ represents the mineral particle density and is set at 2700 kg/m³, while $\rho_{soc}$ is the bulk
density of organic matter, maintained at 130 kg/m³. Also, $m_{soc}$ and $m_g$ denote the percentages of organic matter
and gravel within the topsoil layer.

There are many parameters in equations 7–17. Some parameters have already been assessed for their

importance in ET estimation by Yuan et al. (2021). There are too many studies on investigating the empirical
paramters in equation 7–13. We will not repeat these analysis again. The parameterization method of $\theta_{sat}$ in the
estimation of $r_s$ in this study is composed of various empirical parameters ($\rho_p$, $\rho_{soc}$, and $\theta_{sat, sc}$) for different soil
types. We have conducted a uncertainty analysis of the estimated $\theta_{sat}$ and sensitivity of its uncertainty to the
changes of empirical parameters. The impact of the empirical parameters on the estimation of ET is illustrated
in Figure A1 (a–c). The results indicate that with a 20% uncertainty range in the estimated parameters $\rho_p$, $\rho_{soc}$,
$\theta_{sat, sc}$ for $\theta_{sat}$, the loss in estimating ET is only below 3%. Thus, the conclusion is drawn that the estimation of
ET is not sensitive to uncertainty in the three parameters. Figure A2 also shows the accuracy of the estimated
$\theta_{sat}$ by the method used in this paper. Additionally, a sensitivity analysis is conducted on the empirical
parameters $a$ and $b$ for calculating $r_s$. Keeping $\theta_{sat}$ and SM constant, $r_s$ exhibits exponential changes with
variations in $a$ and $b$, leading to significant fluctuations in the estimation of ET. Within a 20% range of variation
in $a$ and $b$, the maximum loss in ET exceeded 50% in Figure A1 (d–e). Therefore, it is essential to perform
significance tests on the fitting results of the empirical parameters $a$ and $b$, as well as independent validation of
the final ET estimates. The performance of soil surface resistance $r_s$ estimated by the MOD16-STM model at
the site scale is demonstrated in Fig.3. The observations at the ten stations show that the soil surface resistance
exponentially decreases with the increasing $SM/\theta_{sat}$. The MOD16-STM has caught this exponential law. It has
a coefficient of determination ($R^2$) higher than 0.34, which may enable the model to estimate the TP ET
reasonably.
It demonstrates that the parameterized method of $\theta_{sat}$ maintain a high level of consistency with the
observed values. The sensitivity test show that the factors that have a significant impact on $r_s$ and ET are the
topsoil moisture and soil organic matter content. Figure A3 present the impact of soil organic matter content on
$\theta_{sat}$ and ET estimation at different soil types. Hereby, we have collected the most updated soil texture and soil
moisture data to estimate the soil evaporation resistance.
Table 1. Parameters $a$ and $b$ were based on different soil textures used to calculate surface soil resistances.

| Texture | $r_s = exp\left(a + b \times \dfrac{SM}{\theta_{sat}}\right)$ | |
|---|---|---|
| | $a$ | $b$ |
| Sandy Loam | 7.65 | -7.3 |
| Sand | 5.89 | -8.17 |
| Loamy Sand | 8.02 | -17.37 |
| Silt Loam | 7.09 | -3.79 |
| Loam | 6.82 | -4.33 |

| | | |
|---|---|---|
| Clay Loam | 10.17 | -7.43 |
| Sandy Clay Loam | 9.46 | -4.52 |
| Clay | 10.02 | -6.68 |
| Silty Clay | 11.67 | -7.25 |
| Silty Clay Loam | 8.93 | -9.14 |


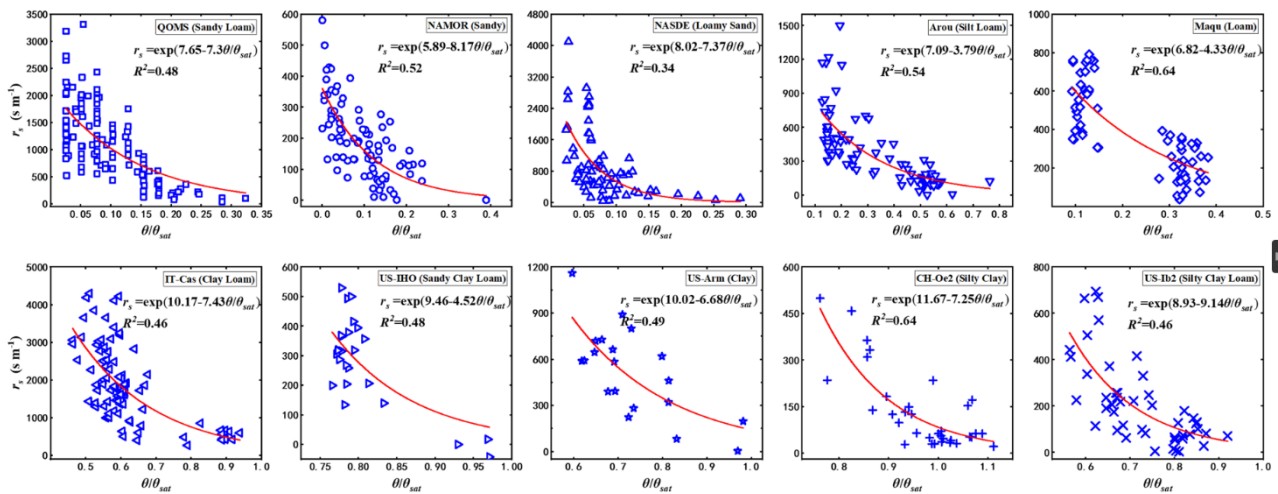


Figure 3. The scatter point relationship between soil surface resistance ($r_s$) and $SM/\theta_{sat}$ observed at QOMS (sandy loam), NAMOR (sandy), NASDE (loamy sand), Arou (silt loam), Maqu (loam), IT-CAS (clay loam), US-IHO (sandy clay loam), US-Arm (clay), CH-Oe2 (silty clay), and US-Ib2 (silty clay loam). The red curves show the equations used by the MOD16-STM model. The site information is given in Table 3 and A1.

**2.2.2 Input data for calculating the TP ET**

The MOD16-STM model relies on various remote sensing datasets, reanalysis datasets, and meteorological forcing datasets to estimate monthly ET across the TP. Specific datasets are carefully selected to minimize spatial and temporal gaps in the final product (Table 2). Here's a breakdown of the critical datasets and their sources:

- Monthly meteorological forcing data, including *wind*, $T_a$, air specific humidity ($q$), $P_a$, *SWD*, and *LWD*, were obtained from the China Meteorological Forcing Dataset (CMFD) with a 0.1° spatial resolution from 1982–2018. This data source was accessed from the National Tibetan Plateau Data Center (Yang et al., 2010; He et al., 2020). CMFD can be downloaded from TPDC (https://data.tpdc.ac.cn/).
- *LST* and precipitation data were sourced from ERA5-Land, which provides data with a spatial resolution of 0.1° and a monthly temporal resolution. These datasets were obtained from the European

Centre for Medium-Range Weather Forecasts (ECWMF).
• Albedo ($\alpha$) data, with a spatial resolution of 0.05° and an 8-day temporal resolution, were derived from
the Global Land Surface Satellite (GLASS) dataset (Liang et al., 2021).
• A long-term NDVI dataset, with a spatial resolution of 0.05° and daily temporal resolution, was
acquired from the National Oceanic and Atmospheric Administration's National Centers for Environmental
Information (NOAA-NCEI) (https://www.ncei.noaa.gov/products/climate-data-records/normalized-
difference-vegetation-index). This dataset calculates the canopy height and Leaf Area Index (LAI) (Chen
et al., 2013).
• A topsoil moisture dataset for the 0–10 cm depth, with a spatial resolution of 0.25° and a monthly
temporal resolution, was obtained from the Global Land Evaporation Amsterdam Model (GLEAM)
(Miralles et al., 2011).
• Upward longwave radiation ($LWU$) was derived from LST using the Stefan-Boltzmann Law. The
emissivity of mixed pixels was calculated based on the specific emissivity values for vegetated and bare
land surfaces, following Sobrino et al. (2004).
• Soil texture and soil property data were obtained from the Harmonized World Soil Database v1.2
(HWSD) (Wieder et al., 2014). These data were used to calculate soil evaporation resistance.
Daily and 8-day input data were averaged over the temporal scale to create monthly datasets to ensure
consistency. The average value was considered invalid if the ratio of valid data in any given month was below
90%. Additionally, the spatial resolutions of all input datasets were interpolated to a standard 0.05° spatial
resolution using a widely used bilinear interpolation method.
Table 2. Input datasets are used to calculate the ET on the Tibetan Plateau.

| | Data source | Temporal resolution | Availability | Domain | Spatial resolution | Method |
|---|---|---|---|---|---|---|
| $SWD$ | CMFD | 3 h | 1979–2018 | China land | 0.1° × 0.1° | Reanalysis |
| $LWD$ | CMFD | 3 h | 1979–2018 | China land | 0.1° × 0.1° | Reanalysis |
| $T_a$ | CMFD | 3 h | 1979–2018 | China land | 0.1° × 0.1° | Reanalysis |
| $q$ | CMFD | 3 h | 1979–2018 | China land | 0.1° × 0.1° | Reanalysis |
| $wind$ | CMFD | 3 h | 1979–2018 | China land | 0.1° × 0.1° | Reanalysis |
| $P_a$ | CMFD | 3 h | 1979–2018 | China land | 0.1° × 0.1° | Reanalysis |
| Precipitation | CMFD | 3 h | 1979–2018 | China land | 0.1° × 0.1° | Reanalysis |

| | | | | | | |
|---|---|---|---|---|---|---|
| *LST* | ERA5 | Monthly | 1981–2021 | Global | 0.1° × 0.1° | Reanalysis |
| *α* | GLASS | 8 days | 1981–2019 | Global | 0.05° × 0.05° | Satellite |
| NDVI | AVHRR | Daily | 1981–2019 | Global | 0.05° × 0.05° | Satellite |
| *SM* | GLEAM | Monthly | 1979–2019 | Global | 0.25° x 0.25° | Reanalysis |
| Soil Properties | HWSD | / | / | China land | 0.083°/1 km | / |

**2.3 Validation methods**
**2.3.1 Model validation at site scale**
Limited stations on the TP make it impossible to collect ET observations at all kinds of soil textures. We
have collected datasets from 17 flux sites outside the TP (Table A1 in Appendix A). Five sites are used to verify
the relationship between soil surface resistance and $SM/\theta_{sat}$. This result is presented in Fig. 3. The other twelve
verification sites include ten different soil textures (sandy loam, sand, loamy sand, silt loam, loam, clay loam,
sandy clay loam, clay, silty clay, and silty clay loam) and three surface cover types (grassland, evergreen forest,
and cropland) (Table A1). These twelve sites are used to do model validation at the site scale. When evaluating
the MOD16-STM at the site scale, the meteorological forcing data comes from the station measurement. This
helps us to minimize the simulation uncertainty due to the errors in the model forcing datasets. This methodology
can allow us to diagnose the model's limitation in representing the evapotranspiration process. Figure A4 shows
that MOD16-STM can capture the ET variations at the twelve sites. Table A2 also lists the statistical values of
the daily ET estimation. Since these sites include all kinds of soil textures and different canopy covers, we
believe the MOD16-STM model could be applied to the TP regional scale.
**2.3.2 ET product evaluation**
The remote sensing ET product is validated through comparison with flux tower observations on the TP.
Table 3 lists details of nine flux stations on the TP used for the ET product evaluation. These stations belong to
the China-Flux (Dang-Xiong site (DX), Hai-Bei site (HB), Yu et al., 2006; Zhang et al., 2019a), the Tibetan
Observation and Research Platform (TORP) (BJ, NADORS, SETORS, QOMS, NAMORS, and Shuang-Hu
(SH), Ma et al., 2020), and the Heihe Watershed Allied Telemetry Experimental Research (HiWATER) (Arou,
Liu et al., 2011, 2018; Che et al., 2019) networks. The nine stations are in areas with three different land cover
types: alpine meadow, alpine steppe, and Gobi. Half-hourly flux data measured by eddy-covariance from the
nine stations are collected. It's important to note that the energy balance closure ratio (ECR) indicates whether
the sum of sensible heat ($H$), latent heat ($LE$), and soil heat flux ($G_0$) matches the $R_n$. Half-hourly data are
screened and corrected accordingly to ensure the reliability of eddy-covariance measurements. Half-hourly $LE$
data is corrected using the Bowen ratio energy balance correction method (Chen et al., 2014).

$$ECR = \frac{H + LE}{R_n - G_0} \tag{18}$$

$$LE_{cor} = \frac{1}{ECR} \times LE \tag{19}$$


Table 3. Details of the nine flux observation stations on the TP used for the ET product evaluation

| Sites | Long., Lat. | Land cover type | Elevation (m) | Availability | Climate zone | Reference |
|---|---|---|---|---|---|---|
| Shuang-Hu (SH) | 88.83°E, 33.21°N | Alpine meadow | 4947 | 2013–2018 | Semi-arid | Ma et al. (2015b) |
| BJ | 91.90°E, 31.37°N | Alpine meadow | 4509 | 2010–2016 | Semi-arid | |
| NADORS | 79.60°E, 33.38°N | Alpine steppe | 4264 | 2010–2018 | Arid | |
| SETORS | 94.73°E, 29.77°N | Alpine meadow | 3326 | 2007–2018 | Sub-humid | Ma et al., 2020 |
| QOMS | 86.95°E, 28.35°N | Gobi | 4276 | 2007–2018 | Semi-arid | |
| NAMORS | 90.99°E, 30.77°N | Alpine meadow | 4730 | 2008–2018 | Semi-arid | |
| Arou | 100.46°E, 38.05°N | Alpine meadow | 3033 | 2008–2017 | Sub-humid | Liu et al., 2011, 2018; Che et al., 2019 |
| Dang-Xiong (DX) | 91.06°E, 30.49°N | Alpine meadow | 2957 | 2004–2010 | Semi-arid | Yu et al., 2006; |
| Hai-Bei (HB) | 101.32°E, 37.61°N | Alpine meadow | 3190 | 2002–2010 | Sub-humid | Zhang et al., 2019a |


In the validation process, the half-hourly $LE_{cor}$ data obtained from all the flux sites are subjected to further

processing, including conversion to daily and monthly averages, while employing a stringent quality control
procedure. Daily values are null if derived from valid data points amounting to less than 80% in a single day.
Similarly, monthly average values are disregarded in the validation if they are derived from valid data points
accounting for less than 80% of observations for that month. This approach ensured the robustness of the
validation process.
**2.4 Accuracy metrics**

The accuracy of the modeled ET was assessed by comparing the pixel values ($M_i$), corresponding to the

latitude and longitude of the flux site, with the flux tower measurements ($G_i$). Several statistical metrics are
employed for validation, including the Coefficient of Determination (R²), a measure of the proportion of the
variance in the observed data ($G_i$) that is explained by the modeled data ($M_i$). A higher R² value indicates a
stronger linear relationship between the two datasets. Mean Bias (MB) represents the average difference
between the modeled ET ($M_i$) and the observed flux tower measurements ($G_i$). Positive MB values suggest
overestimation by the model, while negative values indicate underestimation. Root Mean Square Error (RMSE)
measures the standard deviation of the differences between modeled and observed values ($M_i-G_i$). A smaller
RMSE implies greater accuracy in the model's predictions. Index of Agreement (IOA) indicates the degree of
agreement between modeled and observed data, with a value of 1 indicating perfect agreement. Higher IOA
values indicate better agreement between the two datasets. The equations for these parameters are as follows:

$$R^2 = \frac{\left(\sum_{i=1}^{n}(M_i - \overline{M})(G_i - \overline{G})\right)^2}{\sum_{i=1}^{n}(M_i - \overline{M})^2 \sum_{i=1}^{n}(G_i - \overline{G})^2}, 0 \leq R^2 \leq 1 \tag{20}$$

$$MB = \frac{1}{N}\sum_{i=1}^{n}(M_i - G_i) \tag{21}$$

$$RMSE = \sqrt{\frac{1}{n}\sum_{i=1}^{n}(M_i - G_i)^2} \tag{22}$$

$$IOA = 1 - \frac{\sum_{i=1}^{n}(M_i - G_i)^2}{\sum_{i=1}^{n}(|M_i - \overline{G}| + |G_i - \overline{G}|)^2} \tag{23}$$

The subscript $i$ denotes individual samples, and $n$ is the total number of samples used in the assessment. The
significance of each parameter helps evaluate the model's performance in estimating ET.

**3. Results**
**3.1 Evaluation of ET products against flux tower measurements**

The reliability of remote sensing-based ET estimates is often questioned without ground measurement

verification. This study compares the simulated monthly ET rates from the 0.05° grid where each EC site is
located with the flux tower observational data to validate the MOD16-STM ET results. The validation outcomes
for monthly MOD16-STM ET, using flux tower data, are illustrated in Fig. 4. The modeled ET exhibits excellent
performance and high consistency across the TP compared to ET observations.

Specifically, the grassland sites (SETORS, Arou, DX, and HB) display strong agreement, with R² and IOA

values exceeding 0.82 and 0.95, respectively. The NAMORS site has the lowest performance, with the highest
RMSE (17.8 mm/month) and the lowest $R^2$ and IOA (0.63 and 0.87, respectively). On average, the mean $R^2$ and
IOA values exceed 0.83 and 0.93, respectively. All $R^2$ values pass the significance test at the $p<0.05$ level. The
mean |MB| and RMSE values are less than 3 mm/month and 14 mm/month. It's important to note that positive
MB values indicate an overestimation of ET, particularly during the dry season over barren land (QOMS, DX,
SH, and NADORS) (Fig. 4). Conversely, underestimation occurs at higher ET rates in the summer, likely
because the soil is close to saturation, leading to an overestimation of $r_s$ and underestimation of $E_s$ and ET.
Generally, the time series of ET variations at the nine flux tower stations exhibit seasonal and annual periodic
variations (Fig. 5). The site-scale validation results demonstrate that MOD16-STM ET provides accurate
estimates in the TP region.

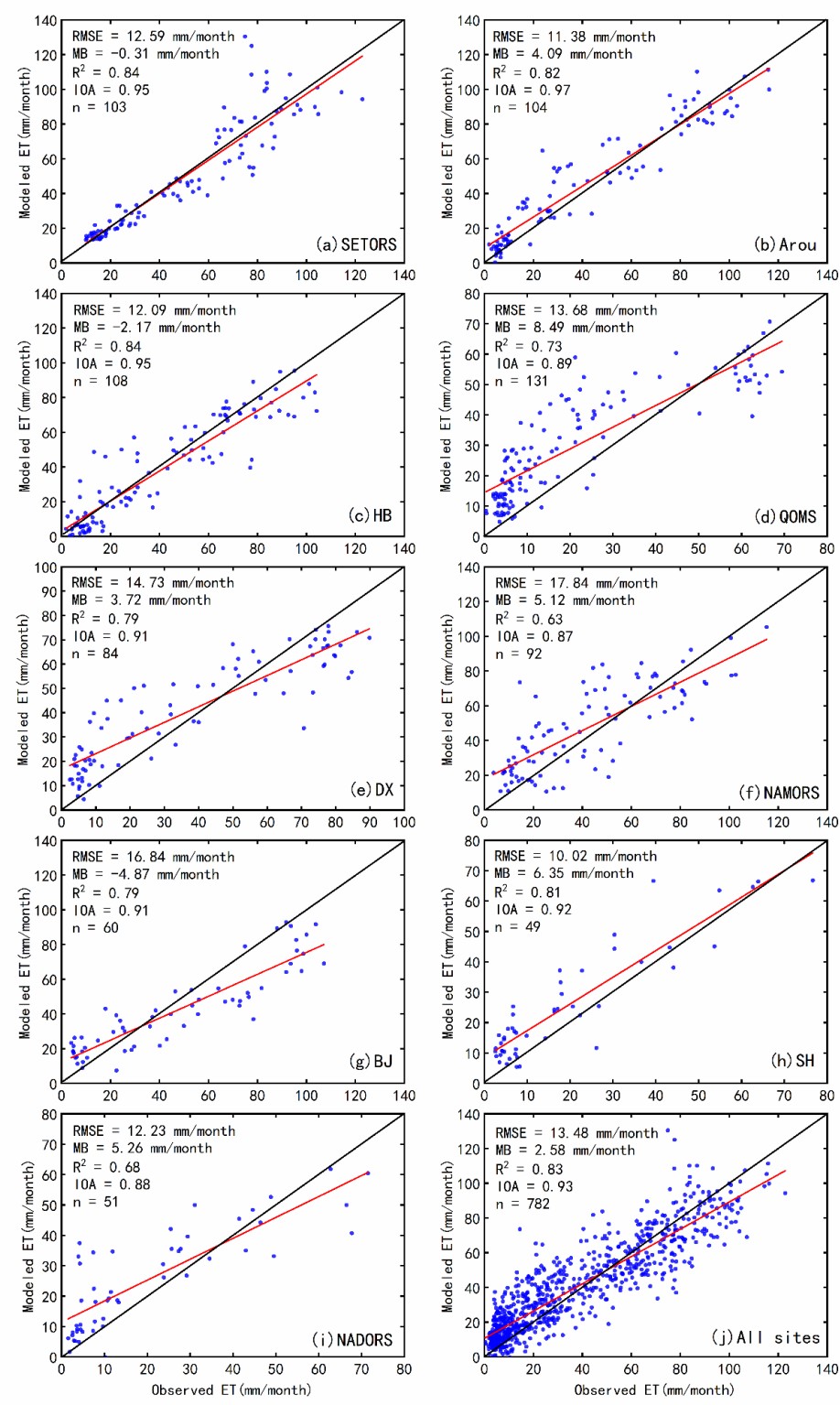


Figure 4. The validation of the MOD16-STM monthly ET at (a) SETORS, (b) Arou, (c) HB, (d) QOMS, (e) DX, (f) NAMORS, (g) BJ,
(h) SH, (i) NADORS, and (j) all sites.

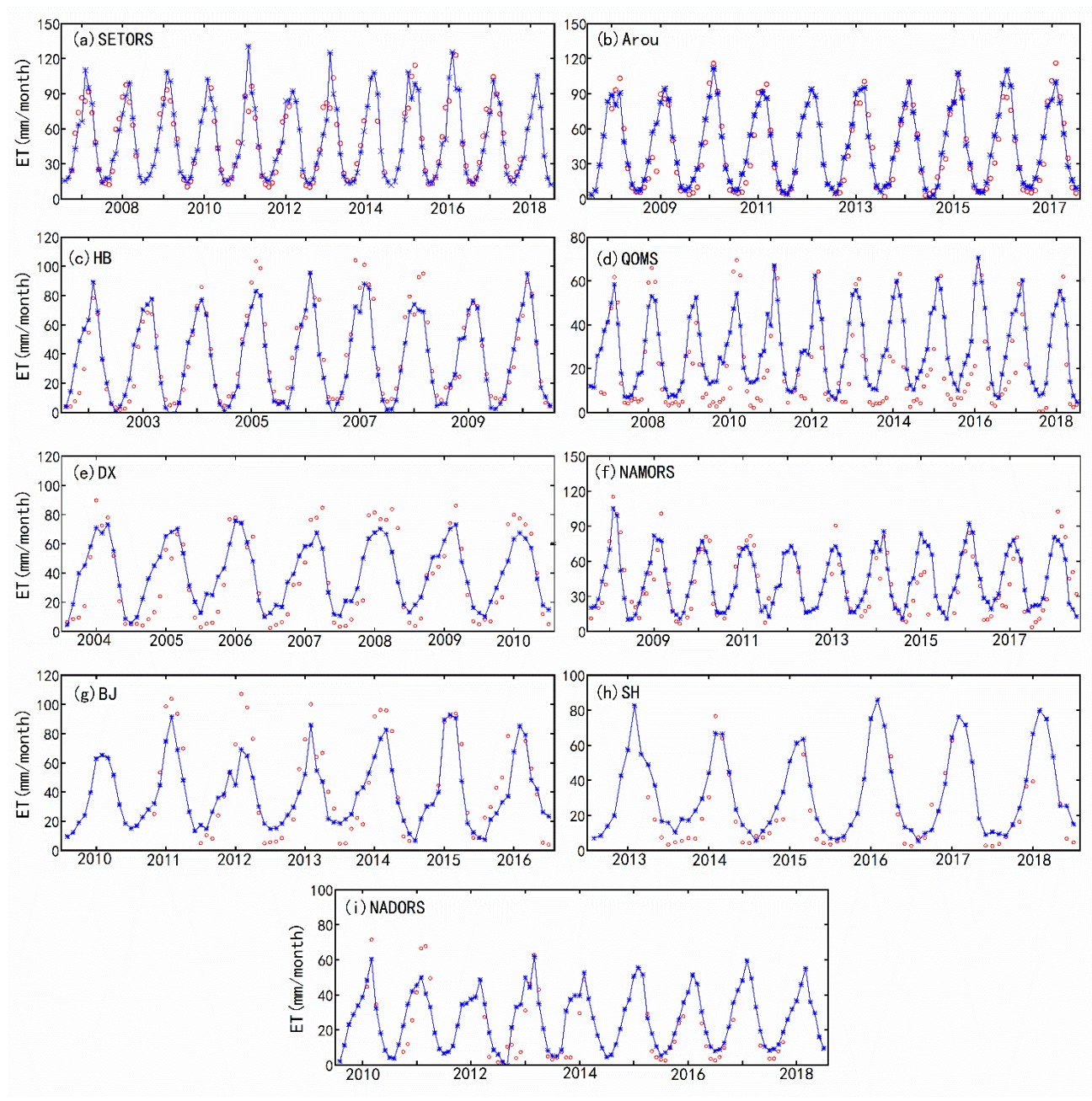

Figure 5. Time series variations in the MOD16-STM simulated ET (blue solid line with "*" marks) and flux-tower-observed ET (red circles) at (a) SETORS, (b) Arou, (c) HB, (d) QOMS, (e) DX, (f) NAMORS, (g) BJ, (h) SH, and (i) NADORS.

## 3.2 Spatial pattern of the multiyear averaged ET across TP

Figure 6 displays the spatial distribution of the average annual ET and its three components across the TP. ET exhibits a decreasing trend from southeast to northwest, with the highest values exceeding 1000 mm/year in the southeastern TP (the Heng-duan Mountains) and the lowest values of less than 100 mm/year in the Qaidam Basin and northwestern TP. This spatial pattern of annual ET closely mirrored that of the aridity index (Fig. 1b),

which is influenced by atmospheric demand and water supply. The sub-humid zone, covering approximately 32.9% of the TP, contributes the highest proportion (43% of the TP's total ET) compared to other climate zones. $E_s$ dominates the central and western TP, with its spatial distribution closely resembling the overall ET. The spatial distributions of $E_c$ and $E_w$ are in line with the distribution of vegetation. High values of $E_c$ (>200 mm/year) and $E_w$ (>50 mm/year) are primarily concentrated in densely vegetated areas such as the Heng-duan Mountains in the southeastern TP.

The multiyear average ET for each season on the TP is depicted in Figure 6, covering spring (March, April, and May), summer (June, July, and August), autumn (September, October, and November), and winter (December, January, and February). The estimated ET reflects the general seasonal patterns quite accurately. During spring, the average ET is higher than in autumn, ranging from 20 to 250 mm/month in spring and from 20 to 150 mm/month in autumn. This difference can be attributed to the increase in surface water generated by the thawing of permafrost and snow and ice melting as temperatures rise in spring, intensifying surface evaporation processes. Additionally, vegetation transpiration increases during the growing season. In summer, ET exceeds 200 mm/month over most TP, except for large areas in the northwestern TP where ET remains below 100 mm/month. Conversely, in winter, lower ET values are observed primarily in the densely vegetated southeastern region of the TP due to reduced water availability (precipitation) and lower $T_a$ across the entire TP during this season.

Over the TP, the multiyear seasonal ET averages across the entire TP are as follows: 90.79±3.16 mm/year (0.23±0.0081 Gt/year) in spring, 152.05±8.44 mm/year (0.38±0.021 Gt/year) in summer, 71.96±2.86 mm/year (0.18±0.0074 Gt/year) in autumn, and 30.54±1.85 mm/year (0.077±0.0047 Gt/year) in winter. The multiyear average ET is 346.5±13.2 mm/year, representing both the mean and standard deviation, which characterizes interannual variability. This corresponds to approximately 0.88±0.034 Gt/year. Among its components, $E_s$ accounted for 292.36±10.39 mm/year (0.74±0.027 Gt/year), $E_c$ amounted to 47.85±3.34 mm/year (0.12±0.006 Gt/year), and $E_w$ is 7.07±2.89 mm/year (0.02±0.001 Gt/year). Notably, $E_s$ constitutes the majority of ET on the TP, exceeding 84%. Wang et al. (2020) accurately estimated that the water evaporated from all plateau lakes is 0.0517 Gt/year. Therefore, utilizing the area-weighted average method, the annual average water evaporation across the entire TP is approximately 0.93±0.037 Gt/year. Furthermore, the TP has an average annual rainfall of about $1.8 \times 10^3$ Gt/year, estimated by Jiang et al. (2022). Approximately 53% of the TP's precipitation returns to the atmosphere through ET.

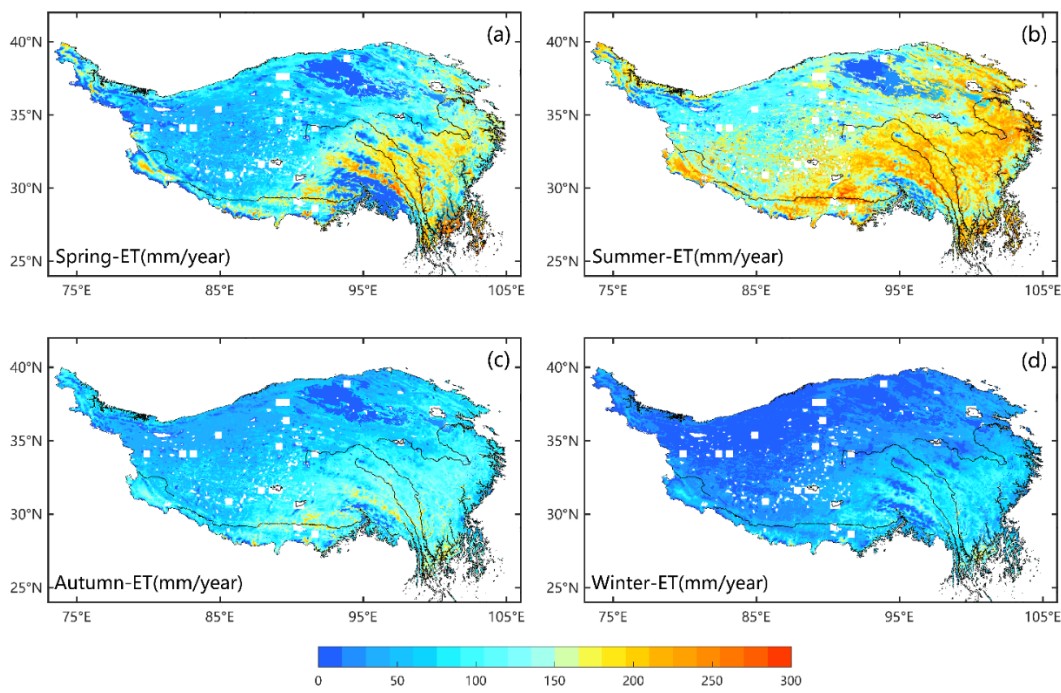


Figure 6. Spatial distributions of the multiyear (1982–2016) mean seasonal ET in (a) Spring, (b) Summer, (c) Autumn,
and (d) Winter across the TP.

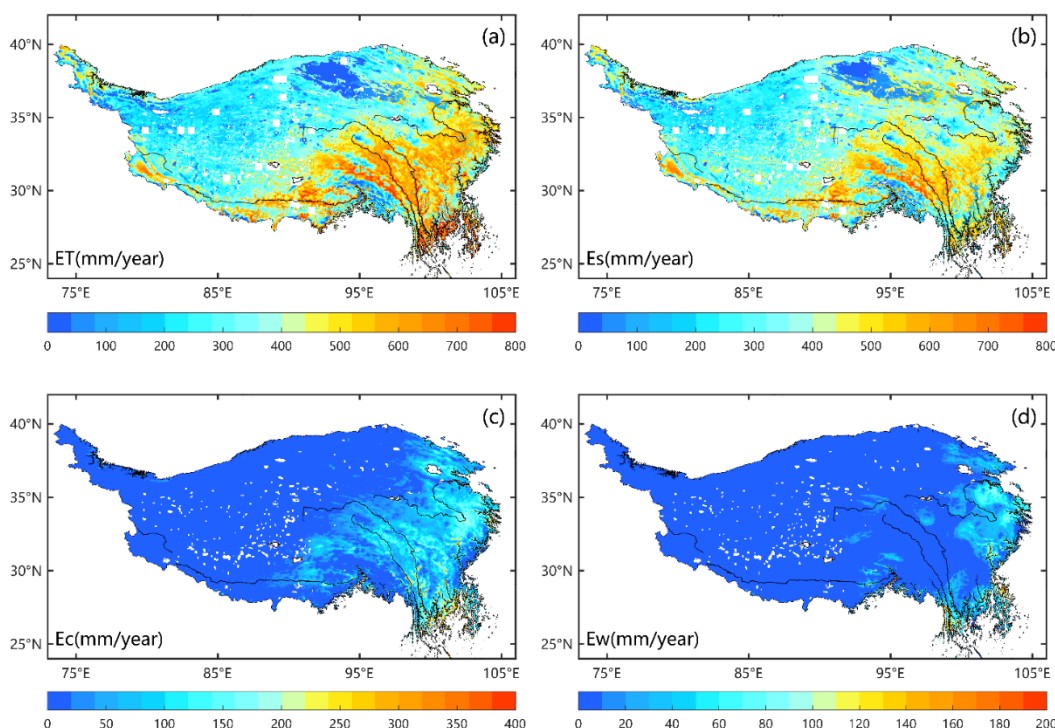


Figure 7. Spatial pattern of the multiyear (1982–2018) mean annual (a) ET (evapotranspiration), (b) $E_s$ (soil evaporation),
(c) $E_c$ (canopy transpiration), and (d) $E_w$ (intercepted water evaporation) across the TP.

## 3.3 Temporal variations in ET across TP

Quantifying variations in ET, both inter- and intra-annual, holds significant importance in understanding monsoon phenomena and studying climate change patterns on the TP. Figure 7 presents the spatial distribution of annual ET and its component trends from 1982 to 2018. These trends exhibit spatial heterogeneity across the TP. The annual ET has seen a significant increase, with rates ranging from 1 to 4 mm/year ($p<0.05$), primarily in the central and eastern TP, encompassing more than 86% of the TP. Conversely, there has been a notable decrease in annual ET, with rates ranging from −3 to −1 mm/year in the northwestern TP. Similarly, the trends for $E_s$ mirror those of ET, albeit with slightly lower magnitudes (1–3 mm/year, $p<0.05$). $E_c$ and $E_w$ have shown slightly increasing trends of 0–2 mm/year ($p<0.05$). When averaged across the entire TP, ET, $E_s$, and $E_c$ exhibited significant increases during the period from 1982 to 2018, with rates of 0.96 mm/year, 0.64 mm/year, and 0.44 mm/year, respectively ($p<0.05$; see Fig. 8). Seasonally, positive, and significant trends are observed in all seasons for ET (Fig. 9), with the strongest trends occurring in summer (0.46 mm/year). Furthermore, multisource ET products indicate that most regions of the TP have exhibited consistent ET changes over the past 30 years (Yin et al., 2013; Peng et al., 2016; Wang et al., 2018; Ma et al., 2019; Wang et al., 2020; Li et al., 2021; Ma et al., 2022).

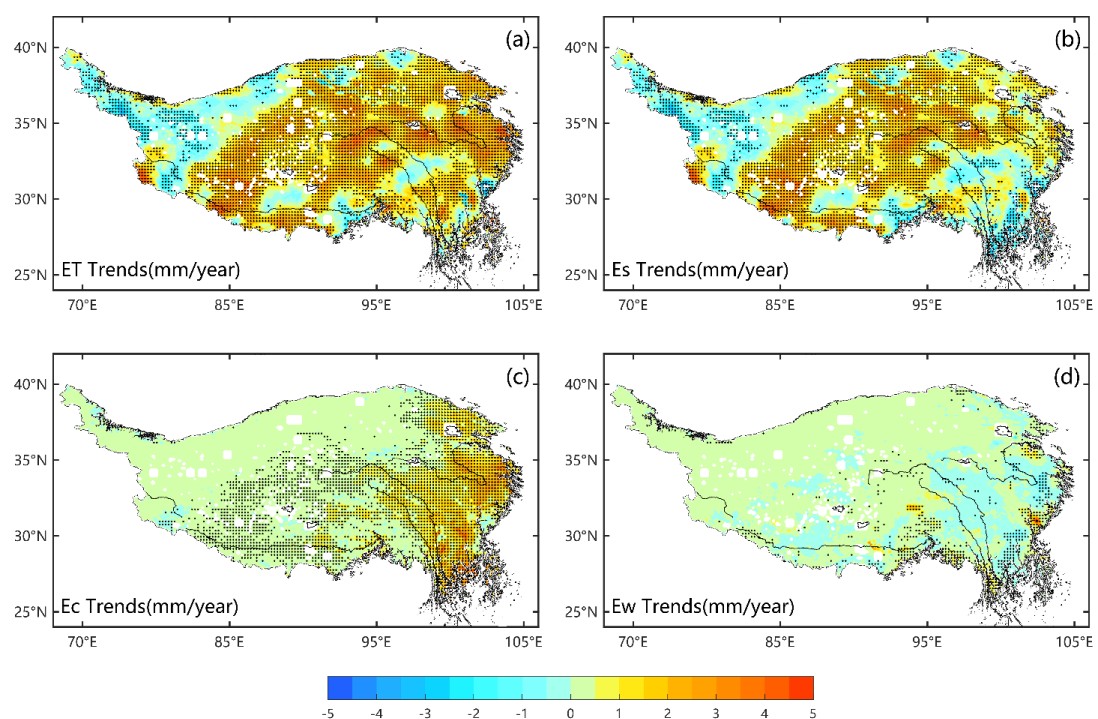

Figure 8. Spatial patterns of the trends (1982–2018) of the annual (a) ET (evapotranspiration), (b) $E_s$ (soil evaporation), (c) $E_c$ (canopy transpiration), and (d) $E_w$ (intercepted water evaporation) across the TP. The stippling on the maps

 indicates the statistically significant trends (*p*<0.05).

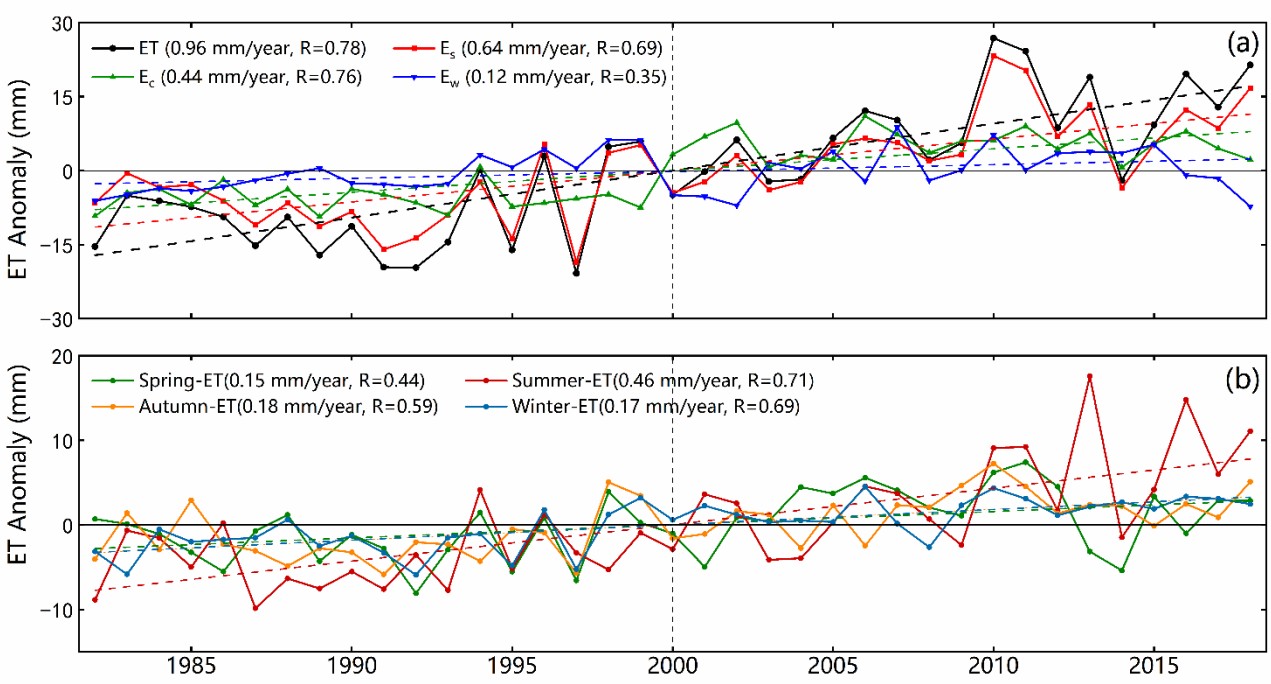

Figure 9. Time series of the (a) anomalies in the annual ET and its three components, and (b) anomalies in seasonal mean ET. The least squares fitted linear trend are demonstrated by the dashed colored lines.

The rise in ET across the entire TP from 1982 to 2018 can be attributed to the concurrent warming and increased precipitation experienced in this region during the same period. Since the 1980s, the TP has undergone a general trend of greening, warming, and heightened precipitation, as illustrated in Figure 10. ET has consistently increased over the past four decades, but there was a notable shift in climate factors around 2000. From 1982 to 2000, ET showed a continuous increase, accompanied by a rapid decline in wind speed, while the $R_n$ remained relatively stable. However, between 2000 and 2018, there was a sharp decrease in $R_n$ alongside an unchanged wind speed, but ET continued to rise during this period.

Consequently, $R_n$ and wind speed are not the dominant factors driving annual variations in ET. The significant increases in $T_a$, *SM*, and precipitation have coincided with the greening of the land surface over the last two decades. These factors collectively contribute to the observed increase in ET. In the most recent decade, the substantial growth in *SM* has emerged as the primary control factor for ET growth.

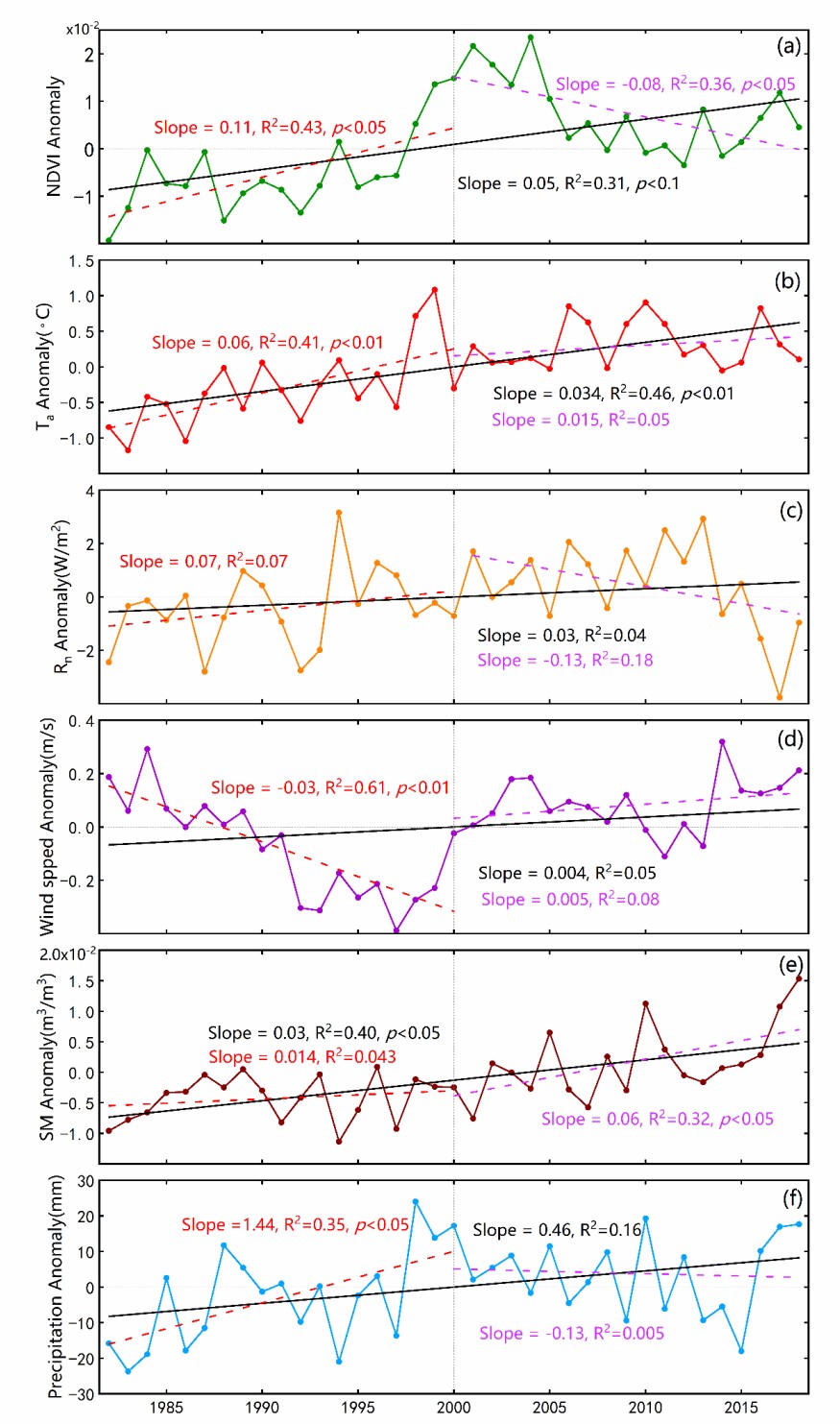

Figure 10. Time series of the annual anomalies in the (a) NDVI, (b) $T_a$, (c) $R_n$, (d) $u$, (e) $SM$, and (f) precipitation and their least squares fitted linear trends during two periods of 1982-2000 and 2000-2018.

In summary, the increase in ET over the TP can be attributed to multiple factors. The rise in available surface water plays a significant role throughout the study. Additionally, there is evidence of a general increase in precipitation across the TP (Fig. 10). The combined impact of warming (shown by $T_a$ in Fig. 10) and

vegetation greening (shown by *NDVI* in Fig. 10) further facilitate the opening of vegetation stomata, promoting increased vegetation transpiration. Warming the land surface and increased wind speeds enhance the efficiency of turbulent water exchange between the land and atmosphere. Furthermore, land surface warming accelerates the melting permafrost and glaciers on the TP. The surface wetting and the thickening of the active soil layer facilitate water transport from the lower soil layers to the upper layers. These environmental changes, such as water availability, precipitation patterns, vegetation dynamics, and temperature trends, all contribute to the increase in ET over the TP.

### 3.4 Comparison of the MOD16-STM product to other ET products over the TP

We have compared the accuracy of the MOD16-STM product and other available TP region datasets. It is shown in Fig. 11. The MOD16-STM ET model demonstrates high performance on the TP, with an average $R^2$ value of 0.87 and an average RMSE of 13.48 mm/month. Wang et al. (2018) evaluated a modified PML model for ET estimation on the TP, reporting $R^2$ values exceeding 0.85 and RMSE values lower than 14 mm/month. The spatially averaged ET for 1982–2012 is 378.1 mm/year. Wang et al. (2020) assessed the performance of the generalized nonlinear complementary principle for ET estimation based on flux tower observations from the TP. Their results indicated an $R^2$ of 0.93 and a RMSE of 0.40 mm/day. The spatially averaged ET during 1982–2014 is 398.3 mm/year. Han et al. (2021) used a combination of the effective aerodynamic roughness length and the surface energy balance model to estimate ET for the entire TP from 2001 to 2018 (Han-ET). They found good agreement between modeled and in-situ measured values ($R^2 > 0.81$, RMSE $< 14.5$ mm/month), and the average annual ET is approximately $496 \pm 23$ mm, which is higher than the $346.5 \pm 13.2$ mm obtained in this study (Fig. 12). The discrepancy can be attributed to differences in models and periods used in the two studies. Ma et al. (2022) also employed the PML_V2 model to estimate ET on the TP (PML), yielding $R^2$ and RMSE values ranging from 0.4 to 0.9 and 0.3 to 0.8 mm/day, respectively. The 35-year mean annual ET rates from PML-Ma resulted in an average value of $353 \pm 24$ mm/year for the entire TP. Notably, the proportion of soil evaporation estimated by PML-Ma was approximately 64% of the total ET, which is lower than the estimated 84% in this study. The primary reason for this difference may be attributed to variations in land cover classification. The MODIS land cover classification largely categorizes the land surface in the northwestern TP as bare soil, increasing the proportion of soil evaporation.

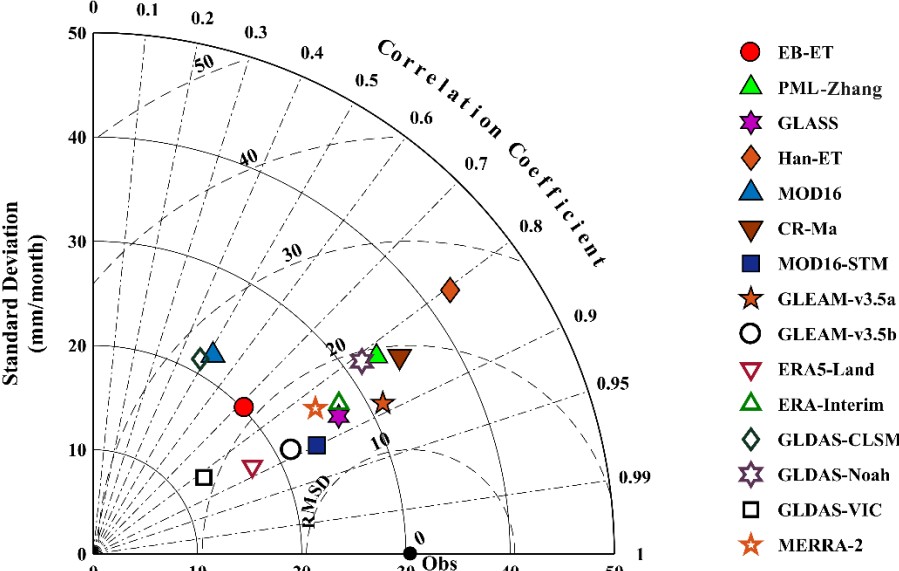


Figure 11. Taylor diagram of the monthly-scale ET dataset validated with flux ET observations.
**4. Discussions**
**4.1 Cross-Comparison of the Spatial Distribution of ET on the TP**

A cross-comparison of the multi-year average values of various ET products is conducted to assess the

differences and consistency in their spatial patterns. From the spatial distribution of annual average ET (Figure
12), all the ET products for the TP exhibit a decreasing trend from southeast to northwest, consistent with the
transition in surface types from forests to grasslands and bare soil. In the Henduan Mountains region, all
products show high values (>600 mm/year) due to the dense vegetation and ample precipitation. However,
significant absolute differences are observed among these 15 ET products. There are high differences among
the products in the sparsely vegetated western and central TP regions. In the central TP region, where the Han-
ET product exhibits the highest annual ET (>600 mm/year), while GLDAS-VIC has the lowest (approximately
35 to 50 mm/year). In the northwestern TP, EB-ET, GLDAS, MERR-2, and GLEAM-v3.5a products display
low values (<50 mm/year), while others range between 100 and 300 mm/year. In the extremely arid Qiangtang
Plateau, all products show low values due to limited available surface water. ERA-Interim, ERA5-Land, PML-
Zhang, CR-Ma, MOD16-STM, and GLEAM-v3.5b have relatively balanced distributions in the central and
western TP regions (200–350 mm/year). There are high differences in the distribution of ET among the products
in the downstream area of the Yarlung Tsangpo River. The spatial resolution of our product is 0.05°. This might
be the reason for MOD16-STM has low ET for this topographic complex region. It is worth noting that MOD16
ET product has many missing values in the northwestern TP region, making it inadequate for a comprehensive
assessment of ET across the entire TP.

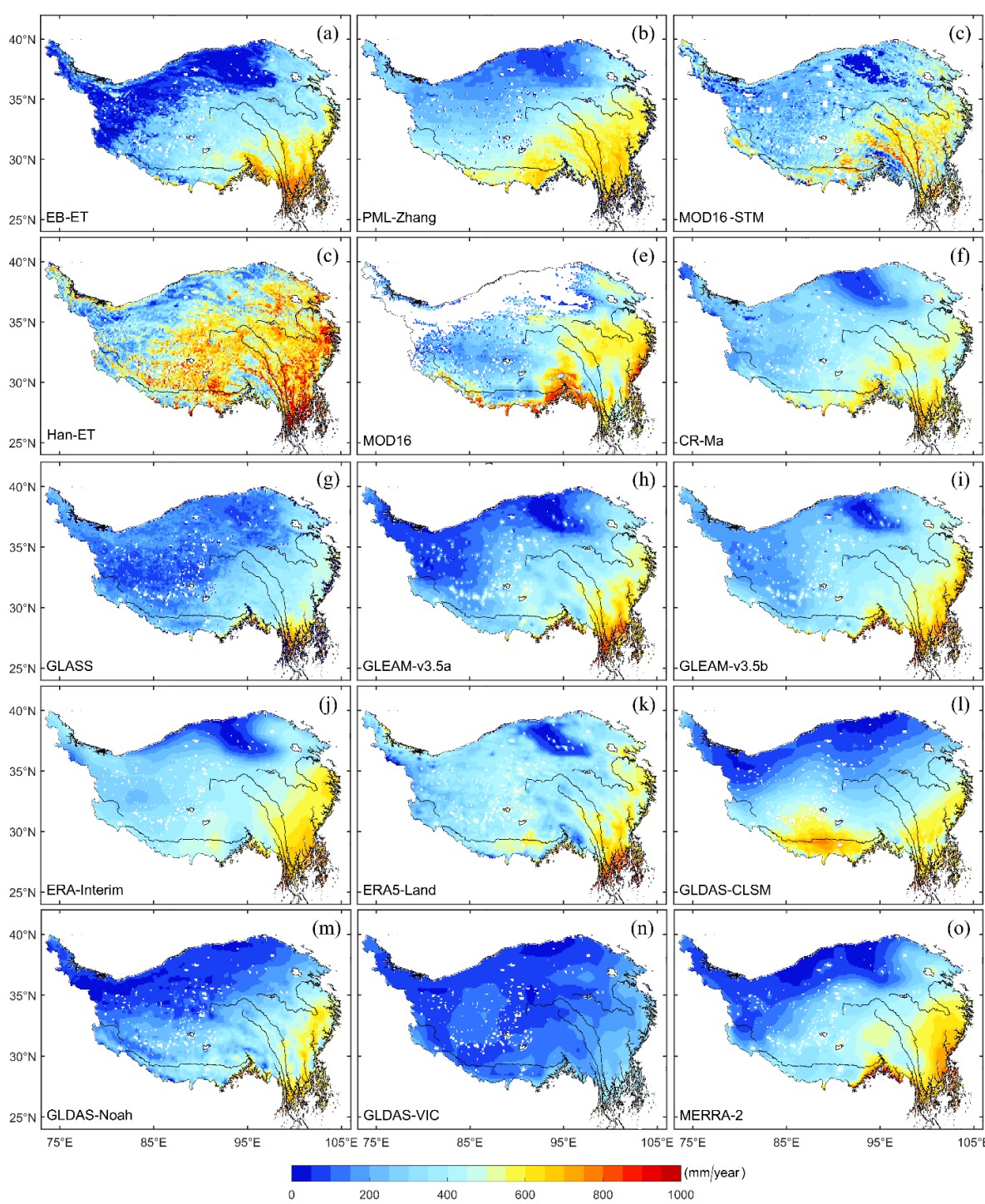

Figure 12. Spatial distribution of annual averaged ET on the TP during 2000 to 2014 derived from 15 products.

**4.2 ET components partitioning**

It is also necessary to have ET components comparative validation to enhance the practicality of the data generated in this study. Unfortunately, there are no measured ET component data publicly available at the moment. Comparative validation can be conducted based on existing research findings. Cui et al. (2020)

estimated the $E_c$/ET at the Nagqu Station (31.37°N, 91.90°E; 4509 m above sea level) in the central region of
the TP using laser spectroscopy and chamber methods. During the observation period, the isotopic-based $E_c$/ET
ranged from 15% to 73%, with an average value of 43%. We calculated $E_c$/ET from our dataset at the same
location and time period (June and July). The values of $E_c$/ET from MOD16-STM are in the range of 13.1% to
62.6%. The average of $E_c$/ET is 38.4%±4.7%, which has a difference of 4.6% relative to isotopic estimation.
Our $E_c$/ET estimation is close to the observation at Nagqu. Guo et al. (2017) also pointed that $E_c$ constituted less
than half of total ET (41% annually, 29% during monsoon) in Magazangbu catchment over the TP.

Moreover, we assess the similarities and differences between MOD16-STM and GLEAM-v3.5a ET

components on the TP. Figure 13 shows that GLEAM's $E_s$ values are generally smaller than our estimation
throughout the TP region. The most recent results from Zheng et al. (2022) also suggest that the GLEAM product
underestimates global $E_s$ outputs. Conversely, GLEAM's $E_c$ values are overestimated in the central and eastern
TP. The differences in $E_w$ are minimal because the values in that region are inherently small. Previous research
has indicated that in the central TP region, $E_s$/ET accounts for over 60% (Cui et al., 2020), and the average
$E_s$/ET ratio across the entire region exceeds 65% (Wang et al., 2018). The reason for the relatively higher $E_s$ in
the central TP is that this region primarily consists of high-altitude grassland as the underlying surface. In the
summer, the dominant processes are $E_c$ and $E_s$, but in the winter $E_s$ becomes the predominant process.
Consequently, the proportion of $E_s$ is higher over the entire year. GLEAM's results show that the $E_c$ process
predominates in the central TP, which differs somewhat from the findings of this study. Zheng et al. (2022) also
indicate that the $E_s$ process predominates in the central TP, exceeding 300 mm/year. Therefore, the ET
components in this study, when compared with previous research, are more in line with the actual conditions in
the TP.

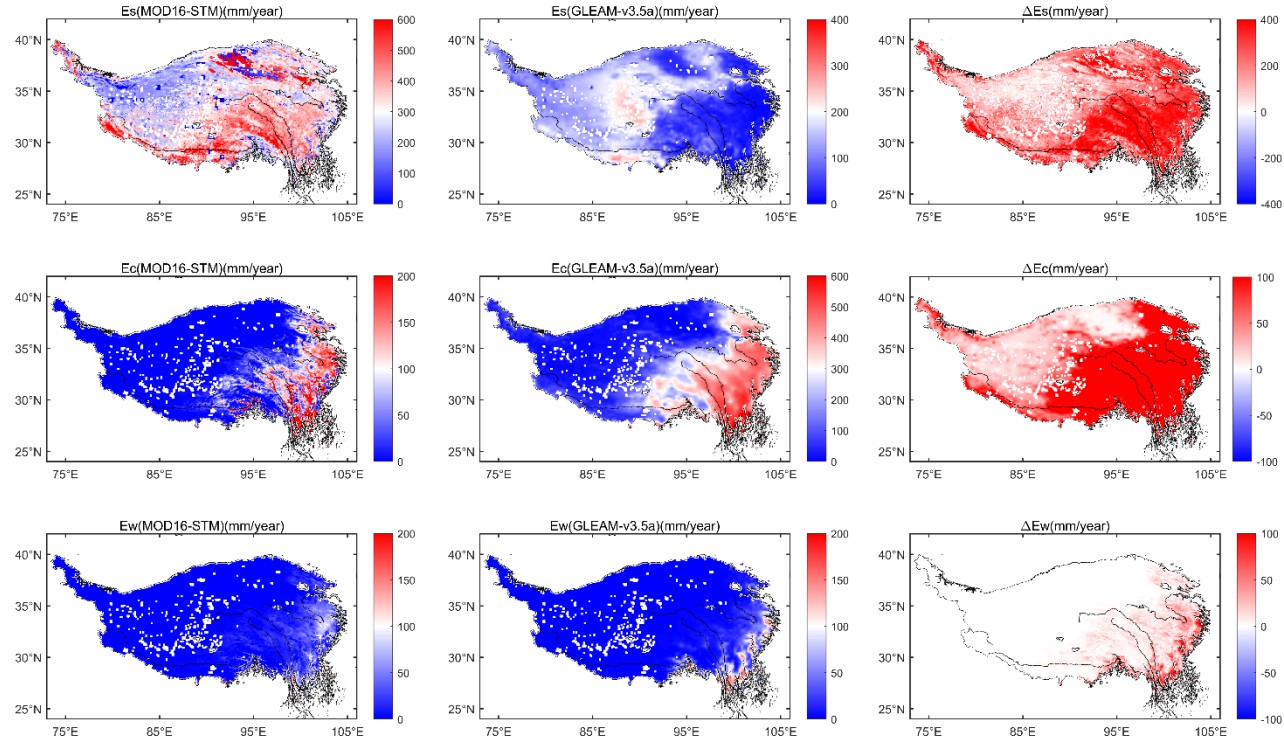

Figure 13. Spatial comparison of ET components and their differences (MOD16-STM minus GLEAM-v3.5a).

**4.3 Discrepancy in the estimation of annual ET over the TP**

Figure 14 provides a comprehensive overview of the periods covered by various ET datasets and their annual ET estimations for the TP. Yao et al. (2013) estimated TP's ET (PT-Yao) in China using a satellite-driven modified Priestley–Taylor algorithm, constrained by NDVI and apparent thermal inertia derived from temperature changes. Their reported mean annual ET for the TP is approximately 320 mm/year. Song et al. (2017) estimated TP's ET (PM-Song) from 2000 to 2010 using the improved Penman–Monteith method and meteorological and satellite remote sensing data at a 1 km spatial resolution. They concluded that the average annual ET on the TP is 350.3 mm/year. Additionally, 18 mean annual ET values are estimated using existing ET products (PML-Zhang (Zhang et al., 2019b), EB-ET (Chen et al., 2019), CR-Ma (Ma et al., 2019), CMIP6-ssp126 (Eyring et al., 2016), GLDAS-Noah (Rodell et al., 2004), GLASS (Liang et al., 2021), GLEAM-v3.5b (Miralles et al., 2011, 2016), ERAR-Land (Muñoz-Sabater et al., 2021), MTE (Jung et al., 2010), PM-Li (Li et al., 2014a, 2014b), LPJ-Yin (Yin et al., 2013)) are included for comparison. Han-ET, ERA5-Land, and CMIP6 produce the highest values (>400 mm/year), while LPJ-Yin, GLASS, EB-ET, GLDAS, and GLEAM values are less than 300 mm/year. The results demonstrate substantial variability in the TP's estimated mean annual ET values. These differences are influenced by objective factors such as data accuracy, limitations of validation

method, and algorithm flaws. The ensemble mean of all datasets yields an annual ET of 348.6 mm/year, with

the MOD16-STM model's estimation (346.5 mm/year) being the closest to this ensemble mean. Overall, the

MOD16-STM ET model again demonstrates acceptable TP performance.

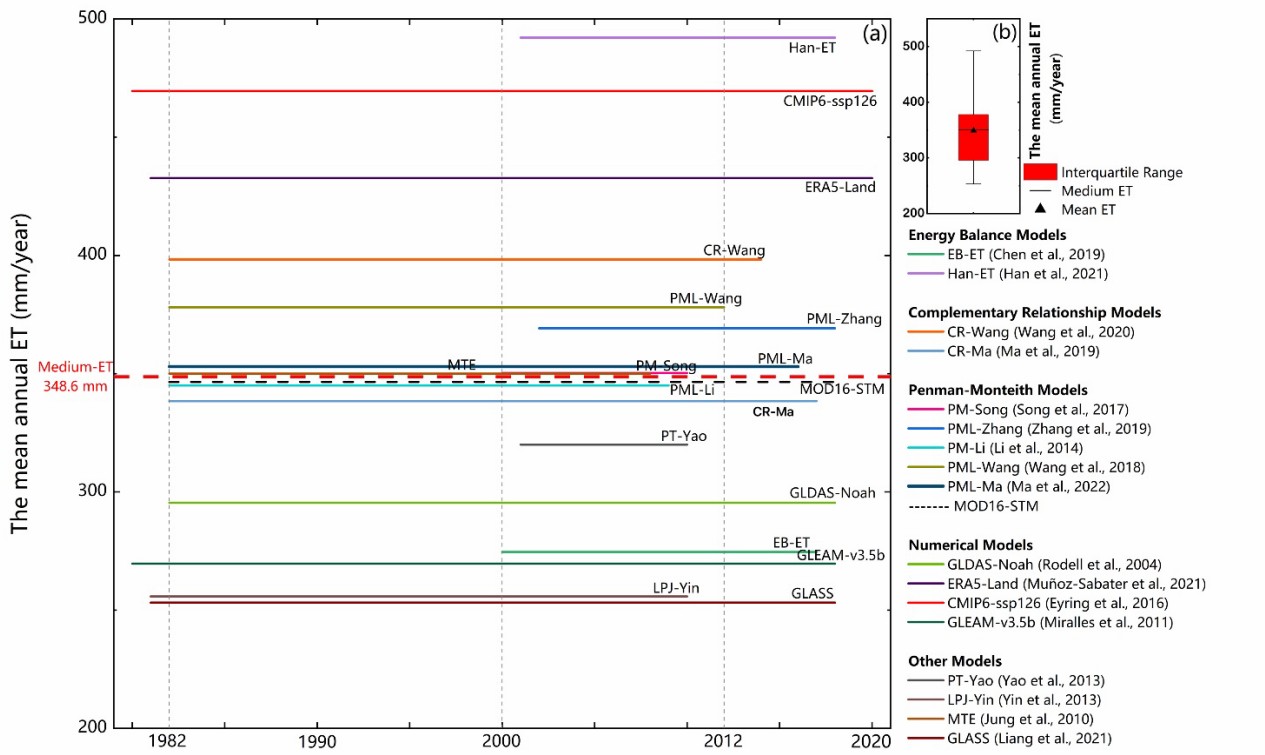

Figure 14. (a) The annual mean ET values of 18 datasets. The *x*-axis is the time coverage of the ET datasets, and the *y*-

axis is the multiyear mean value. (b) The bars denote the mean values and variations of the annual ET.

**4.4 Errors caused by objective factors**

The MOD16-STM and other models rely on remote sensing and reanalysis data as primary input sources.

However, it's essential to acknowledge the inherent uncertainty in these datasets (Ramoelo et al., 2014). For

example, the topsoil water content from satellite data includes some errors (Liu et al. 2021). This indicates that

SM from GLEAM may introduce uncertainties to our ET estimation. Some studies have documented the

greening of the TP. Figure 10a demonstrated a significant decrease in NDVI after 2000, contrasting with the

NDVI changes reported by Wang et al. (2022). This inconsistency highlights the considerable uncertainty in the

NDVI data.

Additionally, *LST* plays a fundamental role in calculating the surface energy balance. Consequently, errors

in ERA5 *LST* can also bring uncertainty to the ET estimation. This study used a threshold value of NDVI (0.25)

to categorize land surfaces as bare soil or canopy-covered pixels. This threshold value may miss-classify bare

soil and grassland on the TP. The land cover mismatches between the reality and the land surface types in the MOD16-STM ET model can also introduce errors in the model simulation.

It is worth noting that flux towers used for validation typically cover areas ranging from a few hundred square meters to several square kilometers. These validation sites' representativeness depends on observation instrument height, turbulence intensity, topography, environment, and vegetation conditions. While site-scale evaluations of the MOD16-STM ET are conducted in this study, it's essential to recognize the uncertainties stemming from the limited number of validation sites. Future research should consider validation across various land cover types, climate zones, elevations, and seasons to provide a more comprehensive assessment of model performance.

While the MOD16 model directly estimates ET, bypassing the need for calculating sensible heat, it still relies on empirical coefficients, particularly those redefined for different soil textures. However, the remaining empirical parameters, such as $C_L$ (the mean potential and stomatal conductance per unit leaf area), can introduce uncertainties into simulation results. Thus, future studies should prioritize parameterizing these empirical factors based on physical processes to reduce simulation uncertainties. It's crucial to consider the influence of physical processes related to deeper soil water and heat transfer on resistance. The MOD16-STM algorithm's accuracy is highly dependent on higher-precision soil moisture products. Since a substantial portion of the TP is covered by permafrost and seasonally frozen soil, assessing soil moisture conditions during freezing and thawing periods becomes challenging. Consequently, it is essential to employ observations during freeze-thaw periods to validate the model's applicability.

In summary, enhancing the model by incorporating physical parameterizations, especially for empirical coefficients, and accounting for the complexities of soil moisture variations in frozen regions will reduce uncertainties in *ET* simulation results in future research.

**5. Conclusion**

In this study, we have developed a 37-year (1982–2018) monthly ET dataset with a high spatial resolution (0.05°) for the TP using the newly developed MOD16-STM model. This dataset covers the entire study area with high spatial resolution and a long time series, making it a valuable resource for climate studies. Then, we investigated ET's spatial distribution and temporal trends across the TP. Key findings are summarized below:

- The ET product generated by the MOD16-STM model exhibits strong performance on the TP. Compared to flux tower observation data, the model achieves high $R^2$ and IOA values of 0.83 and 0.93,

respectively, with an RMSE of 13.48 mm/month and a modest bias (MB) of 2.58 mm/month. This ET dataset holds potential applications in water resource management, drought monitoring, and ecological studies.

- The ET on the TP displays spatial heterogeneity and temporal variations driven by a combination of atmospheric demand and water supply. Generally, annual ET decreases from the southeastern to the northwestern regions of the TP. $E_s$ accounts for over 84% of the annual ET, and the estimated multiyear mean annual ET on the TP for 1982–2018 is 346.5±13.2 mm. This corresponds to an annual water evaporation of about 0.93±0.037 Gt from the entire TP.

- Significant temporal trends are observed in the ET. Most parts of the central and eastern TP exhibit increasing trends of about 1 to 4 mm/year ($p<0.05$), whereas the northwestern TP shows a decreasing trend of −3 to −1 mm/year ($p<0.05$). Averaged across the entire TP, the ET increased significantly at a rate of 0.96 mm/year ($p<0.05$) from 1982 to 2018. This increase in ET over the entire TP from 1982 to 2018 can be attributed to the warming and wetting of the climate during this period.

These findings contribute to a better understanding of the ET dynamics on the Tibetan Plateau and provide a valuable dataset for climate research and related applications.

**Data availability**

The monthly ET dataset presented and analyzed in this article has been released. It is freely available at the Science Data Bank (http://doi.org/10.11922/sciencedb.00020, Y. Ma*, X.Chen*, L. Yuan, 2021) and the National Tibetan Plateau Data Center (TPDC) (https://data.tpdc.ac.cn/en/disallow/e253621a-6334-4ad1-b2b9-e1ce2aa9688f/, http://doi.org/10.11888/Terre.tpdc.271913, L. Yuan, X.Chen*, Y. Ma*, 2021). The dataset is published under the Creative Commons Attribution 4.0 International (CC BY 4.0) license.

**Author contributions**

YMM, LY, and XLC led the writing of this paper and acknowledge responsibility for the experimental data and results. LY, XLC, and YMM drafted the document, and LY led the consolidation of the input and simulation dataset. XLC revised the manuscript. This paper is written in cooperation with all the co-authors.

**Declaration of Competing Interest**

The authors declare that they have no known competing financial interests or personal relationships that could have appeared to influence the work reported in this paper.

**Acknowledgments**

We are grateful for the datasets provided by the China-Flux (http://www.chinaflux.org/), Ameri-Flux (https://ameriflux.lbl.gov/), GHG-Europe (http://www.europe-fluxdata.eu/ghg-europe), the National Tibetan Plateau Data Center (https://data.tpdc.ac.cn/zh-hans/data), the European Centre for Medium-Range Weather Forecasts (ECWMF) (https://www.ecmwf.int/), NOAA-NCEI (https://www.ncei.noaa.gov/products/climate-data-records/normalized-difference-vegetation), the Global Land Evaporation Amsterdam Model (https://www.gleam.eu/), and the National Earth System Science Data Sharing Infrastructure (http://glass-product.bnu.edu.cn/). The authors would like to thank all their colleagues at the observation stations on the TP for their maintenance of the instruments.

**Financial support**

This study is funded by the Second Tibetan Plateau Scientific Expedition and Research (STEP) Program (2019QZKK0103 and 2019QZKK0105) and the National Natural Science Foundation of China (42230610, 91837208, 41975009).

**Appendix A: MOD16-STM model validation at flux site out of the Tibetan Plateau**

Table A1. Basic information about the five test sites (which are used to test the relationship between soil surface resistance ($r_s$) and $SM/\theta_{sat}$ in the MOD16-STM model) and 12 verification sites (used for the MOD16-STM model evaluation at site scale). All the stations are located outside of the Tibetan Plateau.

| | Site | Lat; lon | Land cover | $\theta$ (cm) | $f_{sand}$ | $f_{clay}$ | $m_{soc}$ (%) | $\theta_{sat}$ | Soil Texture | Reference |
|---|---|---|---|---|---|---|---|---|---|---|
| | IT-Cas | 45.07; 8.71 | CRO | 5 | 0.28 | 0.29 | 2.6 | / | Clay loam | *Denef et al. (2013)* |
| | US-IHO | 36.47; 100.62 | Bare | 5 | 0.58 | 0.28 | / | 0.53 | Sandy Clay Loam | *Lemone et al.* (2007) |
| Test Sites | US-Arm | 36.61; -97.49 | CRO | 5 | 0.28 | 0.43 | 1.5 | / | Clay | *Fischer et al.* (2007) |
| | CH-Oe2 | 47.29; 7.73 | CRO | 5 | 0.095 | 0.43 | 2.8 | / | Silty Clay | *Alaoui and Goetz* (2008) |
| | US-IB2 | 41.84; -88.24 | GRA | 0~15 | 0.106 | 0.29 | 2.4 | / | Silty clay Loam | / |
| | US-Dk1 | 35.97; -79.09 | GRA | 10 | 0.48 | 0.09 | / | 0.52 | Loam | *Novick et al. (2004)* |
| | US-Fwf | 35.45; -111.77 | GRA | 5 | 0.30 | 0.13 | 3.2 | / | Silt Loam | *Dore et al. (2012)* |
| | US-Wkg | 31.74; -109.94 | GRA | 5 | 0.67 | 0.17 | 1.0 | / | Sandy Loam | *Ameri-Flux* |
| Verification sites | CA-Obs | 53.98; -105.11 | ENF | 5 | 0.72 | 0.05 | 4.3 | / | Sandy Loam | *Ameri-Flux* |
| | CA-Ojp | 53.91; -104.69 | ENF | 5 | 0.94 | 0.03 | 2.5 | / | Sand | *Ameri-Flux* |
| | CA-Ca2 | 49.87; -125.29 | ENF | 5 | 0.74 | 0.03 | 3.0 | / | Loamy Sand | *Ameri-Flux* |
| | CA-Ca3 | 49.53; -124.90 | ENF | 5 | 0.39 | 0.20 | 4.9 | / | Loam | *Ameri-Flux* |
| | US-Dk3 | 35.97; -79.09 | ENF | 5 | 0.25 | 0.34 | 2.4 | / | Silt Loam | *Ameri-Flux* |
| | US-Fuf | 35.08; -111.76 | ENF | 5 | 0.31 | 0.35 | 3.9 | / | Clay Loam | *Ameri-Flux* |
| | US-Ib1 | 41.86; -88.22 | CRO | 2.5 | 0.10 | 0.35 | 1.8 | / | Silty clay Loam | *Denef et al. (2013)* |
| | ES-ES2 | 39.28; -0.32 | CRO | 5 | 0.11 | 0.47 | 3.7 | / | Silty Clay | *Kutsch et al. (2010)* |
| | IT-Bci | 40.52; 14.96 | CRO | 5 | 0.32 | 0.46 | 1.5 | / | Clay | *Denef et al. (2013)* |

Table A2. Assessment results of the daily ET (mm/day) simulated by the MOD16-STM model at the 12 verification sites.
The in-situ meteorological observation data drive this simulation.

|  | Sites | $R^2$ ($p<0.05$) | IOA | \|MB\| | RMSE |
|---|---|---|---|---|---|
| Grassland | US-DK1 | 0.71 | 0.91 | 0.27 | 0.74 |
|  | US-Fwf | 0.59 | 0.84 | 0.06 | 0.55 |
|  | US-Wkg | 0.69 | 0.84 | 0.005 | 0.58 |
| Evergreen Forest | CA-Obs | 0.88 | 0.96 | 0.05 | 0.33 |
|  | CA-Ojp | 0.79 | 0.93 | 0.11 | 0.38 |
|  | CA-Ca2 | 0.77 | 0.92 | 0.23 | 0.49 |
|  | CA-Ca3 | 0.79 | 0.94 | 0.02 | 0.44 |
|  | US-Dk3 | 0.79 | 0.92 | 0.51 | 0.87 |
|  | US-Fuf | 0.58 | 0.81 | 0.33 | 0.66 |
| Cropland | US-Ib1 | 0.65 | 0.88 | 0.39 | 1.08 |
|  | ES-ES2 | 0.87 | 0.91 | 0.04 | 0.94 |
|  | IT-Bci | 0.41 | 0.76 | 0.14 | 1.14 |
| Mean | / | 0.72 | 0.89 | 0.18 | 0.68 |










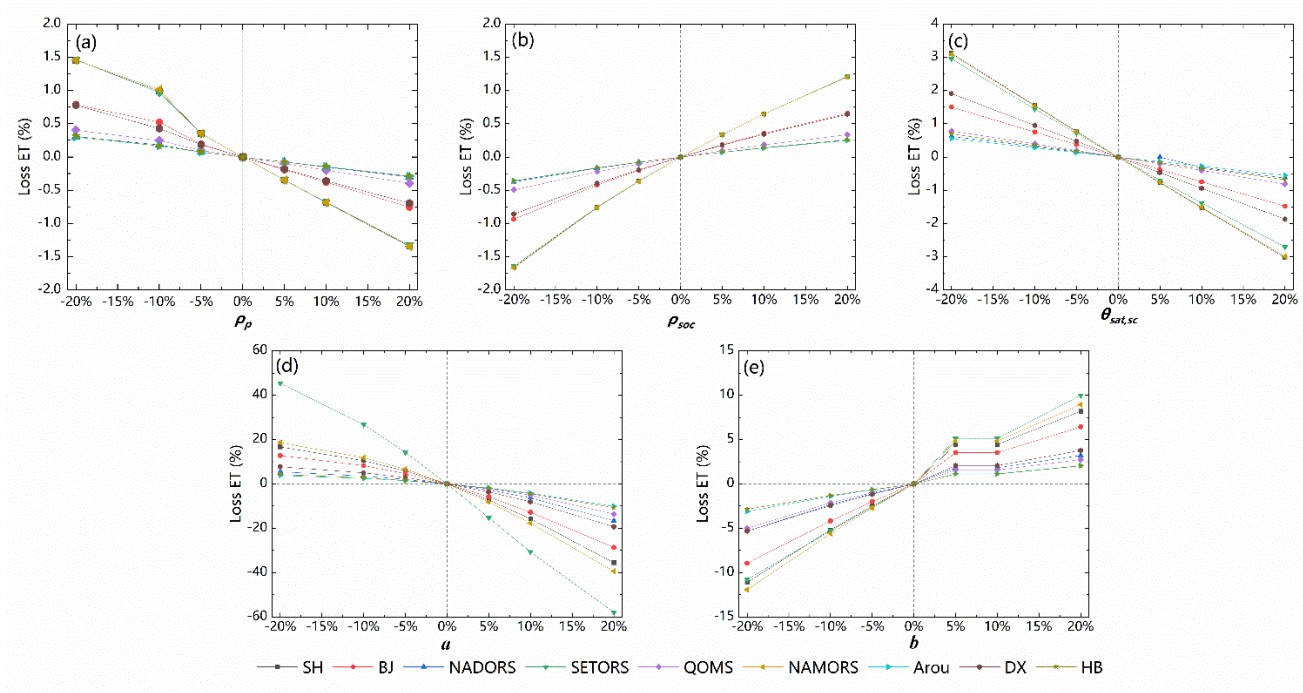

Figure A1. A sensitivity analysis on the impact of the uncertainty in the empirical parameters ($\rho_p$, $\rho_{soc}$, $\theta_{sat, sc}$, $a$ and $b$) to the estimation of ET (test in August, 2018).

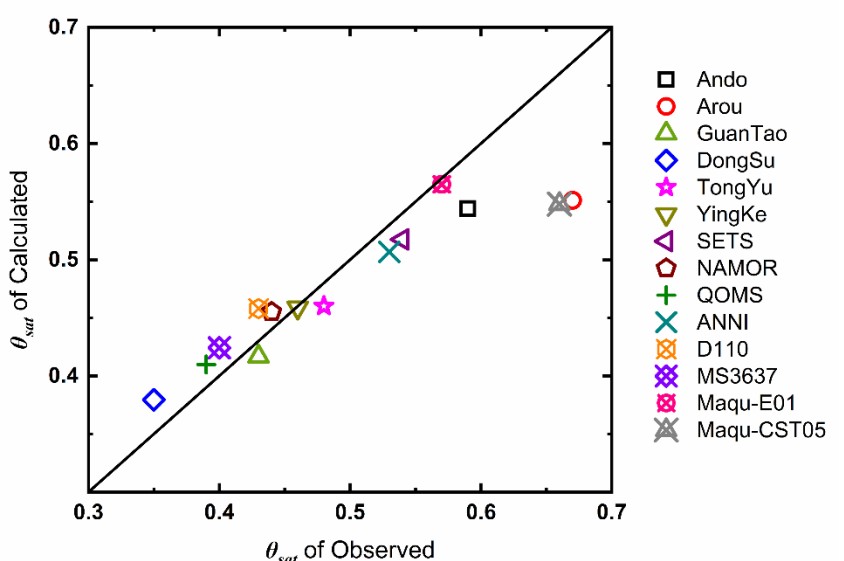


Figure A2. Validation of the consistency between the estimated and the observed values for $\theta_{sat}$ over the TP.














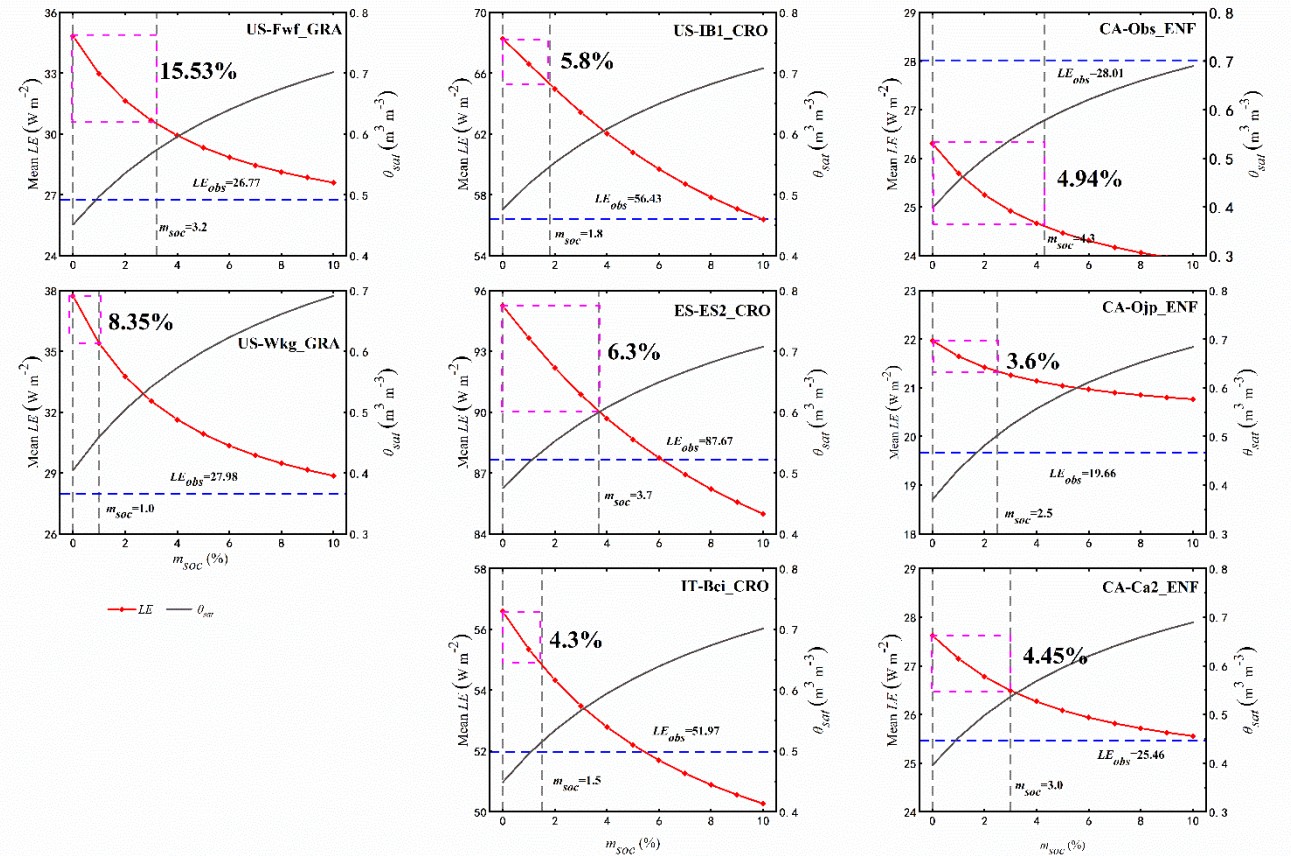

Figure A3. The sensitivity of LE and $\theta_{\text{sat}}$ to the changes of $m_{soc}$ content at different sites..

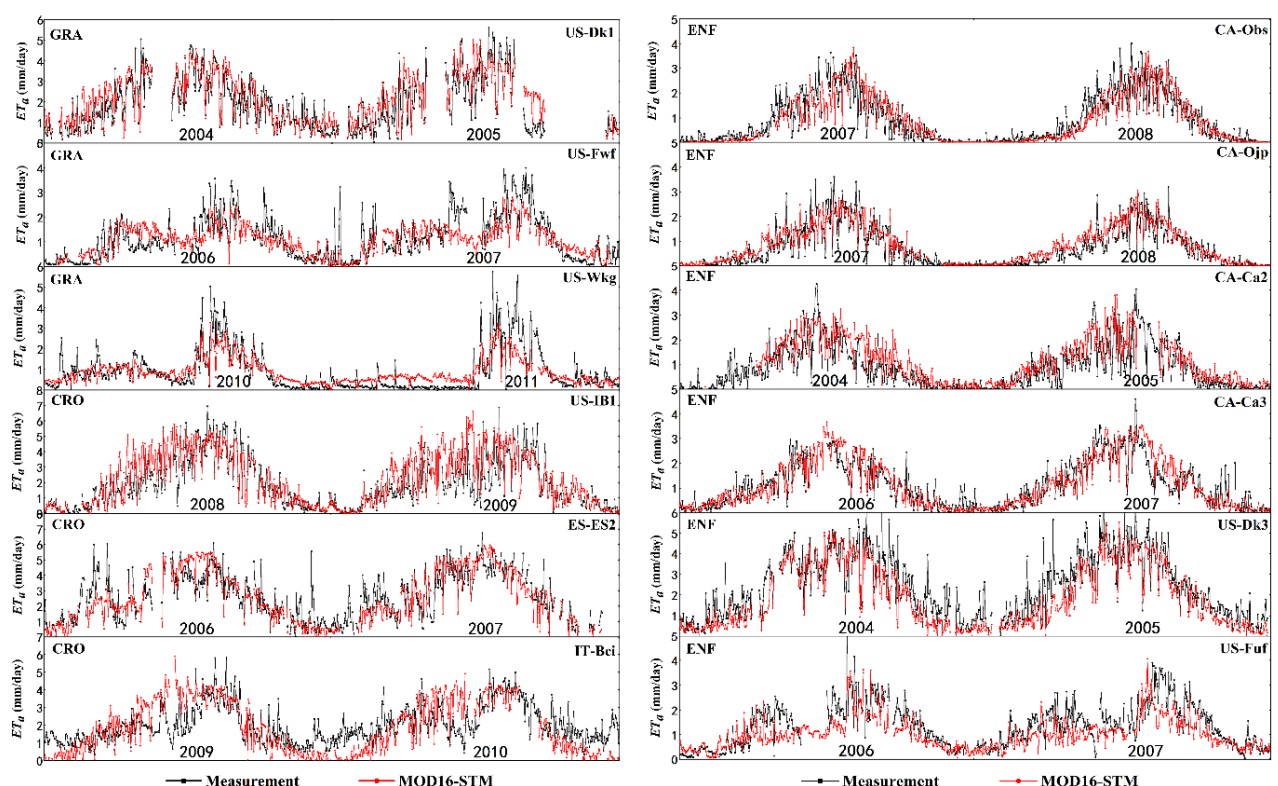

Figure A4. Time-series comparisons of the daily ET estimated by the MOD16-STM model and observations at the 12

verification sites, which include three grassland sites (US-DK1, US-Fwf, and US-Wkg), three cropland sites (US-IB1,

ES-ES2, and IT-Bci), and six evergreen forest sites (CA-Obs, CA-Ojp, CA-Ca2, CA-Ca3, US-DK3, and US-Fuf)

respectively.

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
