# Peer review of "Long-Term Monthly 0.05° Terrestrial Evapotranspiration Dataset (1982–2018)"

_Earth System Science Data, 2022_

## Referee Comment (RC3)

The Tibetan Plateau is the birthplace of Asia's major rivers and is also essential to the Asian energy and water cycles. This manuscript aims to produce a long-term (1982-2018) evapotranspiration product to accurately monitor and understand the spatial and temporal variability of the ET components. The intent of this study is meaningful, and the scope is suitable for publication in Earth System Science Data. However, I also have some concerns about the method and results presented in this study. So I suggest a major revision is needed before publication.

Major comments:

1) The Introduction is generally well written, but the scientific problem of lacking longer-term remote sensing ET products in TP needs to be more evident. It would help readers know why you did this study, the problem of ET products in TP, and how to solve these problems. In addition, the characteristics of TP and required model improvements for TP should be explained in the Introduction.

2) MOD16-STM is utilized in this study, but readers don't know the abbreviation of STM, please give its full name. Three ET components are considered in Eqs. (1) to (3), but the surface resistance rs in Ec, Es, and Ei should be designed reasonably for dry vegetation, wet vegetation, and bare soil, respectively. Eq. (5) is the soil heat flux calculation, but the final term equation should be Is – Ic. Eq. (7) is the calculation of aerodynamic resistance, but the zero plane displacement (d0) is not included in the equation, more details please find in Liu et al. (2007).

Shaomin Liu, Li Lu, Defa Mao, and Li Jia. Evaluating parameterizations of aerodynamic resistance to heat transfer using field measurements. Hydrology and Earth System Sciences, 2007,11(2):769-783.

3) Section 3.4 presented the comparison of MOD16-STM products and other ET products in TP, but readers don't know the spatial pattern difference among different products.

4) The improvements of the proposed model compared with the MOD16 algorithm should be discussed in-depth.

Minor comments:

1) Line 40, 'During the study period, the ET exhibited a significant increasing trend, with rates of about 1–4 mm/year (p < 0.05), over most parts of the central and eastern TP and a significant decreasing trend, with rates of −3 to −1 mm/year, over the northwestern TP'. Please reorganize this sentence.

2) Lines 118-119, how to determine the climate zones, please provide necessary references.

3) Figure 1, all these figures belong to the classification diagram, so the color bar should not be continuous.

4) Line 144, how to calculate Fwet? Are any empirical parameters that need to be calibrated following the studies of Mu et al. (2011, RSE)? How to calibrate these parameters?

Mu, Q., Zhao, M., Running, S.W., 2011. Improvements to a modis global terrestrial evapotranspiration algorithm. Remote Sensing of Environment. 115, 1781-1800.

5) Line 155, what does "surface model" mean?

6) Some grammar mistakes are included, e.g., Line 200, 'domwnload' should be 'download'; Line 330, 'increase' should be 'increased'.

7) Line 210, the URL and necessary references should be added in Table 1.

8) Table 1, the spatial resolution of model input data and auxiliary data is greater than 0.05°. Can the 1km ET products provided in this study capture more spatial details after resampling?

9) Line 225, ECR is calculated in Eq. (18), why not use ECR in Eq. (19) ?

10) Line 231, accuracy evaluation may be more appropriate.

11) Line 265, the legend needs to be added to Figure 4.

12) Lines 292-293, the Es in Ma et al. (2022, AFM) is nearly 226, and Ec is almost 110, please explain the differences.

Ma, N., Zhang, Y., 2022. Increasing Tibetan plateau terrestrial evapotranspiration primarily driven by precipitation. Agricultural and Forest Meteorology. 317.

13) Line 353, are you sure their RMSE values are under 0.006 mm/d?

14) It is quite strange why the NDVI and wind speed have different trends after 2000? Detailed explanations need to be provided into the manuscript.

15) The conclusion is too long to understand, please shorten them to make it more readable.

---

## Author Comment (AC1)

**Responses to Comments Made by Reviewer #1:**

This manuscript generated a long-time evapotranspiration (ET) dataset, including its three components, for the arid and cold areas of the Tibetan Plateau. The intent of the manuscript is worthy and significant, and the topic generally fits the scope of Earth System Science Data. However, I'm afraid the paper still requires thoroughly editing to reach the level of international publications and before publication is granted. One major concern is that the ET estimation methods were not clear enough, i.e., the MOD16-STM was an existing algorithm, what's your contribution? If introducing soil moisture is, how about the estimates without using soil moisture? Furthermore, the validation is somewhat weird. Particularly the components did not perform any validation or the proposed products did not compare to any existing ET products.

Thank you for your suggestion. We have made a thorough editing of the manuscript. Please check the track change word file. Below is our response to the review`s specific comments. We have introduced the contribution of this paper in the line of 75-100. We developed MOD16-STM at site scale in 2021. However, it is not evaluated at continental scale. This paper upscale the application of MOD16-STM from in-situ scale to the TP regional scale. This is the contribution of this paper. Introducing soil moisture is the contribution of our previous paper, Yuan et al. 2021. We have compared the effects of with and without using soil moisture in that paper. We compared with nearly ten other evapotranspiration products in the revised manuscript. The performance of other existing ET datasets were also evaluated and compared, please see Appendix in our revised version. The longer time coverage is an advantage of our ET dataset. Unfortunately, there are no observations of the ET components until now, only EC observations on the Tibetan Plateau can be used to verify the total ET values or soil evaporation values.

[Figure]

**Figure A3** Taylor diagram of the monthly-scale evapotranspiration dataset validated with flux evapotranspiration observations.

**Major concerns:**

- Introduction section. Although the introduction section was well written, it's unclear why the authors perform this study. I would encourage the authors to directly point out the challenges that the present ET products have, rather than stating a lack of long-term remote sensing ET products (which is not true). Give a clear message to the reader what are the critical problems in the studied topic, why you did the study, and what problem(s) will be solved in the current study.

  Thank you for your suggestion. We agree with the reviewer's suggestions. In the introduction we write that: A considerable variance among the ET products for the TP still exhibit (Peng et al., 2016; Baik et al., 2018; Li et al., 2018; Khan et al., 2018). The Penman–Monteith algorithm has also been used to separately estimate the canopy transpiration (Ec), soil evaporation (Es), and canopy intercepted water evaporation (Ew) (Mu et al., 2011; Zhang et al., 2010) for global land. These ET products perform poorly in TP areas with sparse vegetation or arid to semi-arid climates, as well as in areas with inadequate water supplies. The poor performance of the MOD16 Penman–Monteith model (Mu et al. 2011) in the arid to semi-arid areas of the TP is due to the fact that the algorithm does not take into account the dominant role of the topsoil information (topsoil texture and topsoil moisture (SM)) in controlling the evaporation processes (Yuan et al., 2021). Scientists still have

difficulties to accurately separate and validate the ET components on the TP, even though the total ET estimates are consistent across different products (Lawrence et al., 2007; Blyth and Harding, 2011; Miralles et al., 2016).

All the above contents inform readers that challenges of present ET products faces and the critical problems in the TP ET studies. Because, MOD16-STM may provide us with a high chance to accurately estimate ET`s components. That's why we use the model to estimate ET for the TP region.

■ Method section. The authors use a two source PM equation. However, they did not sperate $r_s$ and $r_c$ (Eqs. 1 to 3). In fact, these two resistances as well as resistance for wet canopy (interception) are estimated using different methods in MOD16, but they were estimated in the same way in this study. In addition, the input datasets have different temporal scales and how did you deal with the problem (or model simulation at what kind of temporal scale)? It is also unclear how the estimates were validated. For example, at half-hour or daily scale? How to match the EC tower data with the pixel?

Sorry for the mistake in eq.1. 'rs' in eq 1 should be 'rc'. We have corrected it in the new version. Hereby, we did not estimate resistance in the same way. We parameterized the evaporation resistance $r_s$ in the Es for different soil moisture in the different soil texture (Equation 15). Meanwhile, the estimated CL parameter value of $r_c$ was calibrated at grassland and taken as 0.0038 in the original MOD16 model for Ec over the Tibetan Plateau.

The daily and 8-day model-driven data were averaged over the temporal scale to be monthly datasets. We use pre-processed input data at monthly temporal resolution to calculate the ET.

The results of the long time series simulations are validated by comparing pixels corresponding to the latitude and longitude of flux site with EC measurements. EC hourly measurements have been averaged to monthly values before the evaluation.

■ Results section. I'm somewhat confused by the results shown in figure 5, particularly the ET and its component in forest land. The ET can be high as greater than 700 mm, but the Ec looks like only around 150 mm. Could you show some published data to justify the estimates? Moreover, how accurate is the Ei comparing to the other results (e.g., Zheng's product)? In section 3.4, could you show some

comparisons between your estimates and the other products (e.g., using plots)?

Thank you for your suggestion. There is no forest site on the TP. There is no publicly available data for the observations of ET component. This is why we cannot verify the ET model over the forest. In addition, if you look at the following figure, it shows that the forest land only covers a very small area of TP and most of the forest land is around the margin of the TP border. Hereby, ET of the TP forest is not an important issue. We also validated other existing ET datasets, please check the revised Appendix material.

[Figure]

Figure 1, dark-blue color shows the area of forest land around the southeastern border of TP.

■ Discussion section. In 4.2, insightful discussion is missing. For example, in ET estimation, a lot of empirical coefficients were used, did they cause any uncertainty?

We added the following paragraph to discuss the issues associated to empirical coefficients: Although the MOD16 model directly estimates ET, avoiding the process of calculating sensible heat. The empirical coefficients for the different soil textures were redefined. There are still some empirical parameters (e.g., CL, the mean potential stomatal conductance per unit leaf area) that can increase the uncertainty of the simulation results. Therefore, it is necessary to parameterize these empirical parameters according to physical processes to reduce the uncertainty of the simulation results in future studies. The influence of the physical processes of deeper soil water and heat transfer on the resistance should be considered. The MOD16-STM algorithm has a great dependence on higherprecision soil moisture products. Most areas of the TP are covered by permafrost and seasonally frozen soil. It is difficult to grasp the SM conditions during the soil freezing and thawing period. Therefore, it is necessary to use observations during the soil freeze-thaw period to verify the applicability of the model.

**Specific comments:**

■ Line 68. What did you mean by using "the remote nature of the TP"? Line 77-99. It mentioned that "It is also difficult to separate and validate the ET components effectively." Maybe a comprehensive validation of the ET components is needed to prove that this challenge has been overcome in this dataset. Otherwise, the product does not solve the problem mentioned in the introduction section: "Interestingly, there are significant differences in the global and regional contributions of the Es, Ec, and Ei even if the total ET estimates are consistent across different products (Lawrence et al., 2007; Blyth and Harding, 2011; Miralles et al., 2016)."

(1) "The remote nature of the TP" was changed to 'complex environment of the TP'.

(2) I think the reviewer also agree that a full comprehensive ET components validation is not possible for the TP region. There is no ET components observation at all. In the MOD16-STM model development paper, Yuan et al. 2021, we have innovated a method how to verify and enhance ET components at site scale. We assume that the land cover is bare soil in winter, eddy covariance only observes soil evaporation, hereby, we use winter data to optimize the rs to enhance the soil evaporation equation. Then the enhanced soil evaporation equation was fixed and applied to the summer time. The observed canopy transpiration was assumed to be EC measured ET minus the soil evaporation calculated by the MOD16-STM. The observed canopy transpiration was then used to calibrate $C_L$ and rc. These two steps make us believe that the ET components estimation is reliable.

■ Figure 1. It is better to use different color for each panel. Where are these data from? Thank you for your suggestion. We have modified the caption of figure1 in the revised manuscript as: Figure 1. Maps of the (a) topography (STRM), (b) climate zones (FAO aridity index), (c) land cover types (MCD12C1), and (d) soil textures

(HWSD) in the study area. The red dots indicate the flux site locations.

■ Line 82-83. "The MOD16 algorithm is also used to separately estimate…" may be better.

Thank you for your suggestion. We have modified the sentence in the revised manuscript as: The MOD16 algorithm is also used to separately estimate the canopy transpiration (Ec), soil evaporation (Es), and interception (Ew) (Mu et al., 2011; Zhang   et al., 2010).

■ The authors claim that the new ET product exhibited acceptable performance on the TP based on nine flux towers. Overall, it is agree well with the flux tower ET (Figure 3j), but overestimation occurred at lower ET rates and underestimation at larger ET rates (obviously in Figures 3d, e, f, i). It is better to give some explanation and make insightful discussion (or improvement).

Thank you for your suggestion. The overestimation at lower ET rates may be due to the fact that $r_s$ is underestimated and ET is overestimated. Conversely, underestimation occurred at larger ET rates in summer, probably because the soil was close to saturation and $r_s$ was overestimated leading to an underestimation of Es.

■ Is it necessary to use a question for a section title?

We have modified the sentence in the revised manuscript as: 2.2 Generate a long-term series of monthly ET products

■ I'm quite confused by using the MOD16-STM. What does STM mean (or abbreviation for what)?

Thank you for your suggestion. The full name of STM is "soil texture model". We explain it in the revised manuscript.

■ In the following paragraph, the writing style of T* was varied. Please unify them.

Thank you for your suggestion. "$T*$" was used everywhere in the new manuscript now.

- Although the 18 ET products are proved with accept accuracy, they show a large uncertainty in both trends and averaged ET on the TP. To compare their performances on the TP, it is suggested to compare the averaged ET from these products to the EC measurements (i.e., monthly and annual scales) and water balance method.

  Thank you for your suggestion. We add two new figures to compare the ET products. Please check figure A3, A4 in the Appendix.

- Be careful when using "very", for example Lines 37, 63, 277, 361, etc.

  In the manuscript, we replaced "very" with "quite".

- Figure 4. It is unclear which data is observation and which is observation.

  Thank you. We have modified the caption of figure 4 in the revised manuscript as: Figure 4 Time series variations in the MOD16-STM simulated ET (blue solid line with '*' marks) and flux-tower-observed ET (red circles) at (a) SETORS, (b) Arou, (c) HB, (d) QOMS, (e) DX, (f) NAMORS, (g) BJ, (h) SH, and (i) NADORS.

- Figure 5. What does Ew mean?

  "Ew" means soil and canopy intercepted water evaporation.

- Figure 10. The legends include CR-Ma, while it is not shown in the figure.

  Thank you. This is a mistake. We have indicated which line is CR-Ma in the figure.

- It seems the conclusion is too long.

  Yes, we have made the conclusion part short and concise. Please check the track change file.

---

## Author Comment (AC2)

**Responses to Comments Made by Reviewer #2:**

The manuscript (MS) provides a long-term ET dataset over the Tibetan Plateau, a region that is known as "Asian Water Tower" because it is the source region of a few large Asian rivers. For this reason, accurate information of ET is particularly important. I very much appreciate the considerable efforts made by the authors to the ET community, as shown by not only this gridded dataset but also the eddy-covariance flux observations in Tibetan Plateau. The latter was published by these authors in also ESSD in 2020 (https://doi.org/10.5194/essd-12-2937-2020), which has been widely used by the community to improve the understanding of hydrological and climatological processes in the Tibetan Plateau. The present MS is an obviously big step forward upon the previous one, which is also definitely significant for understanding the land surface processes in the Third Pole Region.

**Minor Revision:**

■ The authors stated their new ET dataset is at 0.01 degree. However, the best resolution of the inputs (Table 1) is just 0.05 degree and this is only for albedo and NDVI, others are even coarser. Usually, for any models (not only ET models), the resolution of model's output cannot exceed the best resolution of inputs, otherwise it becomes simple "resampling" of the data (e.g., nearest neighbor or bilinear interpolation). This does not make sense since it cannot bring new spatial information. Therefore, I think the best resolution of this new ET dataset from their model can only be 0.05 degree.

Thank you for your suggestion. We agree to that resampling does not provide us with more information. Hereby, we fixed our objective from 0.01 degree to 0.05 degree product. Hereby, we have modified the title and other places which mentioned the spatial resolution in the revised manuscript.

■ There have been a great number of studies that reported Tibetan Plateau is greening. This is not surprised because warming and wetting in recent decades may promote vegetation growth in such a cold and dry region. However, the NDVI significantly decreased after 2000 in Fig 9a, while warming (Fig 9b) and wetting (Fig 9e) are

still seen for this period. This seems different with the NDVI reported in Wang et al. (2022). Because NDVI is a key input of the model that determines both canopy transpiration and soil evaporation (fc in the Equations 1 and 2), I would suggest the authors to test if other NDVI datasets (perhaps even other leaf area index data?) also show similar interannual variations and how ET varied if they are also used in the modeling, otherwise the trends in transpiration and soil evaporation in the MS should be interpreted with caution.

Thank you for your constructive suggestions. We found that both the NDVI products from AVHRR and MODIS showed great consistency in time series. AVHRR-NDVI has a longer time series. Therefore, in this study, the AVHRR-NDVI product was used as the model input without obtaining more driver data. Although the existing studies have done a more realistic study of NDVI performance on the Tibetan Plateau, to compare more NDVI product maybe a deviation from the focus of this study. In addition, NDVI significantly decreased after 2000, is because of the extremely high NDVI values in the year of 2001 and 2004. When we remove these two years, the trends from 2000 to 2018 is still increasing. Hereby, the two years NDVI data might not be stable. We will do a deeper study on the influences of different NDVI product in future.

- Ln61: Here the Immerzeel et al. 2010 Science paper should be cited since this paper is perhaps the first one that proposed Asian Water Tower concept?

  We have cited Immerzeel et al. 2010 in the revised manuscript.

  Immerzeel, W. W., Van Beek, L. P. H., Bierkens, M. F. P.: Climate change will affect the Asian water towers. Science, 2010, 328(5984), 1382–1385. https://doi.org/10.1126/science.1183188

- Ln75: Please delete the "pan evaporation" studies here since they are not really relevant to the present topic—ET.

  "Pan evaporation" studies were removed in the new manuscript.

- Ln76-77: Please do not combine EC and reanalysis into one sentence. I would

suggest moving reanalysis to the previous sentence since it belongs to "dataset". However, EC is a kind of ground observation and is much more valuable/reliable than above "datasets", which should be specially highlighted.

Yes, this is a very good suggestion. We have modified the sentence in the revised manuscript as: Some investigations have used reanalysis datasets (Shi et al., 2014; Dan et al., 2017; Yang et al., 2019; De Kok et al., 2020), eddy-covariance measurement (Shi et al., 2014; You et al., 2017; Yang et al., 2019; Ma et al., 2020) to study the ET on the TP.

■ Ln144: Please show how fc is derived from NDVI.

Thanks. We have added the equation in the revised manuscript as:

$$fc = \left(\frac{NDVI - NDVI_{min}}{NDVI_{max} - NDVI_{min}}\right)^2 \tag{4}$$

■ Ln195: Prec is not shown in Tabel 1?

Thank you for your suggestion. We have added the precipitation information in Table1.

■ Ln199-201: Please show the source reference and website of the NDVI dataset.

Thank you for your suggestion. We have added the sentence in the revised manuscript as: A long-term normalized difference vegetation index (NDVI) dataset with a 0.05° spatial resolution and daily temporal resolution were download from the National Oceanic and Atmospheric Administration's National Centers for Environmental Information (NOAA-NCEI) (https://www.ncei.noaa.gov/products/climate-data-records/normalized-difference vegetation-index) and was used to calculated the canopy height and LAI (Chen et al., 2013).

■ Ln205: The emissivity data is not shown in Table 1? Please also show the resolution for it.

Emissivity is calculated following the method of Sobrino et al. (2004). That's why we did not list emissivity data in table 1. Its resolution is 0.05 degree.

Sobrino, J. A., Jiménez-Muñoz, J. C., and Paolini, L.: Land surface temperature retrieval from LANDSAT TM 5, Remote Sens. Environ., 90(4), 434–440, https://doi.org/10.1016/j.rse.2004.02.003, 2004.

- Table 2: I suggest adding the reference for each EC flux observation station in the Table 2. Also, the land cover type is not clear for BJ, does it belong to grassland?

  Thanks. References for each EC flux observatory are added in table 2 now. The land cover type of BJ is the alpine meadow. We have updated all the land covers to alpine meadow and alpine steppe, which is more professional.

- Ln297-298: This point is important, but different precipitation products may produce different ratios of ET to precipitation. Did you test other datasets? I am not fully convinced by the reanalysis Prec product (especially in TP). Please also consider other Prec data,e.g., CMFD, TPHiPr (Jiang et al., 2022), and the latest gauge-satellite merged product GPCP Version 3.2 released in this year, etc. This suggestion also applies to Figure 9f because the lakes in TP continued to rapidly expand after 2000, but Prec from the current Figure 9f even decreased. Please see Fig 4e and Fig 6 in Zhang et al. (2020).

  Thank you for your suggestion. We have used the precipitation product to give a more reliable estimation. The sentence is revised in the manuscript as follows: The average annual rainfall on the TP is about $1.8 \times 10^3$ Gt/year, estimated from the data of ERA5-Land, CMFD, and TPHiPr in Jiang et al., 2022. About 53% of the precipitation on the TP returns to the atmosphere through ET.

- Ln330: What is the difference between "wetting" and "increased precipitation"? The "wetting" occurs many times throughout the MS…. I assume you mean the increased soil moisture? Please state clearly.

  Thank you for your suggestion. We have modified the sentence in the revised manuscript: Since the 1980s, the TP has experienced overall greening, warming, and wetting (increased soil moisture and precipitation).

- Figure 9: Please show the specific periods for different slopes shown in the Figure 9?

  Thanks. There are two periods in the figure. We specify the two slope values for the two time periods in Figure 9.

- Ln430: "evaporated" from the entire TP.

  We have modified the sentence as you suggested in the revised manuscript.

- Ln432: Please add the p value also for the significant decreasing trend.

    Thank you for your suggestion. We have added the p value in the revised manuscript.

- Appendix A: I do not know whether ESSD allows Online Supplementary Materials. If so, I suggest moving all tables and figures in Appendix to Online Supplementary Materials. There is no need to show them (some are even out of TP) in this MS.

    Thank you for your suggestion. It is possible to use Appendix for ESSD journal.

---

## Author Comment (AC3)

**Responses to Comments Made by Reviewer #3:**

The Tibetan Plateau is the birthplace of Asia's major rivers and is also essential to the Asian energy and water cycles. This manuscript aims to produce a long-term (1982-2018) evapotranspiration product to accurately monitor and understand the spatial and temporal variability of the ET components. The intent of this study is meaningful, and the scope is suitable for publication in Earth System Science Data. However, I also have some concerns about the method and results presented in this study. So I suggest a major revision is needed before publication.

**Major comments:**

■ The Introduction is generally well written, but the scientific problem of lacking longer-term remote sensing ET products in TP needs to be more evident. It would help readers know why you did this study, the problem of ET products in TP, and how to solve these problems. In addition, the characteristics of TP and required model improvements for TP should be explained in the Introduction.

Thank you for your suggestion. We have modified the introduction in the revised manuscript to describe the ET products problem and how to solve the problem. The poor performance of the MOD16 Penman–Monteith model (Mu et al. 2011) in the arid to semi-arid areas of the TP is due to the fact that the algorithm does not take into account the dominant role of the topsoil information (topsoil texture and topsoil moisture (SM)) in controlling the evaporation processes (Yuan et al., 2021). TP land covers is dominated by the short and sparse vegetation. Soil moisture may play an important role on the ET estimates for the TP region. The Penman–Monteith algorithm has been used to test the performance of the ET estimation on the TP (Wang et al., 2018; Ma et al., 2022). However, the effects of the SM on the evaporation resistance and stomatal conductance are not included in these studies. The enhanced Penman–Monteith model, MOD16-STM (MOD16 soil texture model), redefines the Es and Ec module to take into account the impacts of SM on soil evaporation resistance, with the help of eddy-covariance observations on the TP (Yuan et al., 2021). Hereby, MOD16-STM may provide us with a high chance to accurately estimate ET`s components.

■ MOD16-STM is utilized in this study, but readers don't know the abbreviation of STM, please give its full name. Three ET components are considered in Eqs. (1) to

(3), but the surface resistance rs in Ec, Es, and Ei should be designed reasonably for dry vegetation, wet vegetation, and bare soil, respectively. Eq. (5) is the soil heat flux calculation, but the final term equation should be Is – Ic. Eq. (7) is the calculation of aerodynamic resistance, but the zero plane displacement (d0) is not included in the equation, more details please find in Liu et al. (2007). Shaomin Liu, Li Lu, Defa Mao, and Li Jia. Evaluating parameterizations of aerodynamic resistance to heat transfer using field measurements. Hydrology and Earth System Sciences, 2007,11(2):769-783.

Thank you for your suggestion. We give the explanation of STM in the new version. We made a mistake in equation 1 which cause the reviewers' misunderstanding. We use different resistance for Ec, Es, and Ei. Please check the revised equations. "Is – Ic" was used in Eq 5 now. Zero plane displacement was added in the equation. Liu et al. 2007 was also cited in the manuscript.

■ Section 3.4 presented the comparison of MOD16-STM products and other ET products in TP, but readers don't know the spatial pattern difference among different products.

Thank you for your suggestion. We added new figure A4 to show the spatial difference among different products. It is also included in the following.

[Figure]

**Figure A4** Spatial distribution of 15 annual mean evapotranspiration on the Tibetan Plateau from 2000 to 2014.

- The improvements of the proposed model compared with the MOD16 algorithm should be discussed in-depth.

  Thank you for your suggestion. We have added the following figure to demonstrate the difference between MOD16 and MOD16-STM in the revised manuscript:

| Symbol | Description | MOD16 | MOD16-SMT |
|---|---|---|---|
| $C_L$ | The mean potential stomatal conductance per unit LAI | 0.007 | 0.0038 |
| $h_c$ | Vegetation height | Not used | In situ |
| $\nu$ | Kinematic viscosity of the air | Not used | $1.328 \times 10^{-5} \times \left(\dfrac{P_0}{P}\right)\left(\dfrac{T_{air}}{T_0}\right)^{1.754}$, $P_0 = 101.3$ kPa and $T_0 = 273.15$ K |
| $u_*$ | Friction velocity | Not used | In situ |
| $L$ | The Obukhov length (m) | Not used | In situ |
| $T_*$ | Friction temperature | Not used | $\dfrac{T_{air}u_*^2}{kgL}$ |
| $z_{0h}$ | Heat roughness length | Not used | $\dfrac{70\nu}{u_*}\exp\left(-7.2u_*^{0.5}\left|T_*\right|^{0.25}\right)$ |
| $z_{0m}$ | Momentum roughness length | Not used | $h_c/8$ |
| $r_s^s$ | Surface resistance of bare soil | $r_{tot}$ | $\exp\left(A + B \times \dfrac{\theta}{\theta_{sat}}\right)$ |
| $r_a^s$ | Aerodynamic resistance of soil surface | $r_a = \dfrac{rh \times rr}{rh + rr}$ | $\dfrac{\ln\left(\dfrac{z_h}{z_{0h}} - \psi_h\right)\ln\left(\dfrac{z_m}{z_{0m}} - \psi_m\right)}{k^2 u}$ |
| $\theta_{sat}$ | Soil porosity | Not used | $\left(1 - V_{SOC} - V_g\right) \times \theta_{sat,m} + V_{SOC} \times \theta_{sat,sc}$ |
| $\theta_{sat,sc}$ | The porosity of the SOC | Not used | $0.489 - 0.00126 \times \%sand$ |
| $V_{SOC}$ | Volumetric content of the SOC | Not used | $\dfrac{\rho_p \times \left(1 - \theta_{sat,m}\right) \times m_{SOC}}{\rho_{SOC} \times \left(1 - m_{SOC}\right) + \rho_p \times \left(1 - \theta_{sat,m}\right) \times m_{SOC} + \left(1 - \theta_{sat,m}\right) \times \dfrac{\rho_{SOC} \times m_g}{1 - m_g}}$ |
| $V_g$ | Volumetric content of gravel | Not used | $\dfrac{\rho_{SOC} \times \left(1 - \theta_{sat,m}\right) \times m_g}{\left(1 - m_g\right) \times \left(\rho_{SOC} \times \left(1 - m_{SOC}\right) + \rho_p \times \left(1 - \theta_{sat,m}\right) \times m_{SOC} + \left(1 - \theta_{sat,m}\right) \times \dfrac{\rho_{SOC} \times m_g}{1 - m_g}\right)}$ |

**Figure A5** Comparison of Improved Parameters or Intermediate Variables in the MOD16-SMT Model and the MOD16 Model (Yuan et al., 2021).

**Minor comments:**

- Line 40, 'During the study period, the ET exhibited a significant increasing trend, with rates of about 1–4 mm/year (p < 0.05), over most parts of the central and eastern TP and a significant decreasing trend, with rates of −3 to −1 mm/year, over the northwestern TP'. Please reorganize this sentence.

  Thank you for your suggestion. We have rewritten the sentence in the revised manuscript as: The results of this study indicate that the ET in most of the central and eastern parts of TP showed a significant upward trend with a rate of about 1-4 mm/year (P<0.05) during the period from 1982 to 2018.

- Lines 118-119, how to determine the climate zones, please provide necessary references.

  The climate zone is based on FAO drought index dataset. We have added the sentence in the revised manuscript as: Based on FAO drought index dataset, it is the largest landform unit in Eurasia and mainly includes hyper-arid, arid, semi-arid, and sub-humid climate zones (Fig. 1b).

- Figure 1, all these figures belong to the classification diagram, so the color bar should not be continuous.

  Thank you for your suggestion. But, to keep consistency with Fig.1a in the figure, Fig.1b-1d these figures belong to the classification figure, and the legend is kept uniform.

- Line 144, how to calculate $F_{wet}$? Are any empirical parameters that need to be calibrated following the studies of Mu et al. (2011, RSE)? How to calibrate these parameters? Mu, Q., Zhao, M., Running, S.W., 2011. Improvements to a modis global terrestrial evapotranspiration algorithm. Remote Sensing of Environment. 115, 1781-1800.

  The detailed equation for $F_{wet}$ and all the other variable in the MOD16-STM model have been listed in Yuan et al. 2021. That`s why we did not repeat it in this paper. $F_{wet}$ was calculated with relative humidity. We did not change it from Mu et al. 2011

$$F_{wet} = \begin{cases} 0.0, RH^4 < 70\% \\ RH^4, 70\% \leq RH \leq 100\% \end{cases} \tag{3}$$

- Line 155, what does "surface model" mean?

  The "surface model" means the 'land surface model'. We changed 'surface model' to 'land surface model'.

- Some grammar mistakes are included, e.g., Line 200, 'domwnload' should be 'download'; Line 330, 'increase' should be 'increased'.

  Thank you for your correction. We have modified the sentence in the revised manuscript.

- Line 210, the URL and necessary references should be added in Table 1.

  Thank you for your suggestion. URL and references for each input data have been included in section 2.2.2 Input data., Such as, CMFD is downloaded from TPDC (https://data.tpdc.ac.cn/). A long-term normalized difference vegetation index (NDVI) dataset with a 0.05° spatial resolution and daily temporal resolution were download from the National Oceanic and Atmospheric Administration's National Centers for Environmental Information (NOAA-NCEI) (https://www.ncei.noaa.gov/products/climate-data-records/normalized-difference vegetation-index) and was used to calculated the canopy height and LAI (Chen et al., 2013).

- Table 1, the spatial resolution of model input data and auxiliary data is greater than 0.05°. Can the 1km ET products provided in this study capture more spatial details after resampling?

  Yes, this is a very good suggestion. Thank you for your suggestion. We have adjusted the spatial resolution accuracy to 0.05° for the optimal data set for this study.

- Line 225, ECR is calculated in Eq. (18), why not use ECR in Eq. (19) ?

  Thanks. Eq. 19 has been revised as you suggested.

- Line 231, accuracy evaluation may be more appropriate.

  Yes, this is a very good suggestion. Thank you for your suggestion. We have modified the title in the revised manuscript.

- Line 265, the legend needs to be added to Figure 4.

  Thank you for your suggestion. We changed the caption of Figure 4 to include legend information. Please check the revision in the revised verion.

- Lines 292-293, the Es in Ma et al. (2022, AFM) is nearly 226, and Ec is almost 110, please explain the differences. Ma, N., Zhang, Y., 2022. Increasing Tibetan plateau terrestrial evapotranspiration primarily driven by precipitation. Agricultural and Forest Meteorology. 317.

  Thank you for your suggestion. The results of this study assume mainly NDVI < 0.25 as a screening condition when classifying bare soil. Therefore, the simulation

results of this study are dependent on NDVI data. The difference with the results of Ma et al. (2022) depends solely on the difference in NDVI-driven data. the NDVI true values used in the model of Ma et al. (2022) have a wider spatial distribution, while the NDVI of AVHRR has a narrow spatial distribution style. This results in why Ec in Ma et al. (2022) is higher than ours.

■ Line 353, are you sure their RMSE values are under 0.006 mm/d?

RMSE values was revied from "0.006mm/d" to "14 mm/month".

■ It is quite strange why the NDVI and wind speed have different trends after 2000? Detailed explanations need to be provided into the manuscript.

Thank you for your suggestion. This is not out of expectation when a different trends exhibit. We agree that the Tibetan Plateau is turning green. The warming and wetting of recent decades may have promoted vegetation growth in such a cold and dry region. The NDVI declined after 2000. This is caused by quite high NDVI values in the year of 2001 and 2004. The short period from 2005-2018 still shows an increasing trend. Therefore, it is possible that 2001 and 2004 has a poor data quality. Wind speed trend changes have been reported by other scientists. The explanation is out of focus of this paper.

■ The conclusion is too long to understand, please shorten them to make it more readable.

Thank you for your suggestion. We have simplified the conclusions of the manuscript, as detailed in the revised manuscript.

1. The ET product generated using MOD16-STM exhibited a good performance on the TP. Compared to the flux tower observation data, the $R^2$ and IOA values of the modeled ET reached 0.83 and 0.93 for 782 samples, with the RMSE of 13.48 mm/month and an MB of 2.58 mm/month. The actual ET and can be used in research in water resource management, drought monitoring, and ecological change.

2. The combined effect of the atmospheric demand and water supply resulted in spatial heterogeneity of the ET and the changes in the ET. The annual ET generally decreased from southeast to northwest on the TP. The Es accounted

for more than 84% of the annual ET. The estimated multiyear (1982–2018) mean annual ET on the TP was 346.5±13.2 mm, resulting in approximately 0.93 ± 0.037 Gt/year of total water evapotranspiration from the entire TP.

3. The ET exhibited a significant increasing trend, with rates of about 1 to 4 mm/year (p<0.05), over most parts of the central and eastern TP and a significant decreasing trend, with rates of −3 to −1 mm/year, on the northwestern TP. Averaged across the entire TP, the ET increased significantly during 1982–2018, with a rate of 0.96 mm/year. The increase in the ET over the entire TP from 1982 to 2018 can be explained by the warming and wetting of the climate during this period.

---

## Author Response (AR2)

**Responses to Comments Made by Reviewers:**

1. Dear authors, Kindly address the 16% similarity issue and ensure proper licensing before we can proceed with reviewer invitation. We kindly ask you to revise your manuscript accordingly and to upload the revised files, a point-by-point reply to the comments, and a marked-up manuscript version showing the changes made no later than 04 Oct 2023 at: https://editor.copernicus.org/ESSD/review-file-upload/essd-2022-195.

   Thank you very much for your feedbacks. Based on the original manuscript, we conducted a similarity check in the paper database and then made revisions. Attached are the revised complete manuscript and the annotated version. We kindly request the reviewer to carefully review them and provide further suggestions.

---

## Author Response (AR3)

**Responses to Comments Made by Editors:**

■ Dear Authors, thank you for your efforts in improving the manuscript. However, one of the reviewers still has critical concerns about the revised version. For instance, since many empirical coefficients were sourced from studies conducted in different regions, it is essential to quantify, evaluate, or discuss the uncertainties associated with these coefficients. Furthermore, as suggested by the reviewer, an evaluation of ET should be included. Moreover, the data description appears to be quite limited. Please thoroughly address these concerns raised by the reviewer in your revised manuscript. Thank you!

Thank you very much for the suggestions provided by the editors and reviewers. We gladly accept them. In this paper, there are numerous empirical parameters in equations 7–17, all of which pertain to the parameterization of $r_a$ and $r_s$ in the MOD16-STM model. These parameters have already been assessed for their importance in ET estimation in the literature by Yuan et al. (2021). There are too many studies on investigating the empirical coefficients in equation 7–13. We will not repeat this analysis again. The uncertainty of $r_s$ to a and b in equation 14 has been reflected by Figure 3 in the manuscript. The parameterization method of $\theta_{sat}$ in the estimation of $r_s$ in this study is composed of various empirical parameters ($\rho_p$, $\rho_{soc}$, and $\theta_{sat, sc}$) for different soil types. We have conducted a uncertainty analysis of the estimated $\theta_{sat}$ and sensitivity of its uncertainty to the changes of empirical parameters. The impact of the empirical parameters on the estimation of ET is illustrated in Figure A1 (a–c). The results indicate that with a 20% uncertainty range in the estimated parameters $\rho_p$, $\rho_{soc}$, and $\theta_{sat, sc}$ for $\theta_{sat}$, the loss in estimating ET is only below 3%. The uncertainty of $\theta$sat to changes in these empirical parameters (equation 15–17) is relatively low. Thus, the conclusion is drawn that the estimation of ET is not sensitive to variations in these three parameters.

We have conducted a uncertainty analysis of the estimated $\theta_{sat}$. Figure A2 shows the accuracy of the estimated $\theta_{sat}$ by the method used in this paper. It demonstrates that the parameterized method of $\theta_{sat}$ could have a good representation of different soil types. Additionally, a sensitivity analysis is conducted on the empirical parameters $a$ and $b$ for calculating $r_s$. Keeping $\theta_{sat}$ and SM constant, $r_s$ exhibits exponential changes with variations in $a$ and $b$, leading to significant fluctuations in the estimation of ET. Within a 20% range of variation in $a$ and $b$, the maximum loss in ET exceeded 50% in Figure A1 (d–e). Therefore, it is essential to perform significance tests on the fitting results of the empirical parameters $a$ and $b$, as well as independent validation of the final ET estimates. The sensitivity test show that the factors that have a significant impact on $r_s$ and ET are the topsoil moisture and soil organic matter content. The following Figure A3 present the impact of soil organic matter content on $\theta_{sat}$ and ET estimation at different soil types. We also add discussions on these sensitivity analysis in the revised manuscript. Please refer to lines 207 to 230 and lines 647 to 675 in the manuscript for more details.

[Figure]

Figure A1. A sensitivity analysis on the impact of the uncertainty in the empirical parameters ($\rho_p$, $\rho_{soc}$, $\theta_{sat, sc}$, $a$ and $b$) of the estimated $\theta_{sat}$ on the estimation of ET (test in August, 2018).

[Figure]

Figure A2. Validation of the consistency between the estimated values and the observed values for $\theta_{sat}$ over the TP.

[Figure]

Figure A3. The sensitivity of LE and $\theta_{sat}$ to the changes of $m_{soc}$ content at different sites.

Chen, Y., Yang, K., Tang, W., Qin, J., and Zhao, L.: Parameterizing soil organic carbon's impacts on soil porosity and thermal parameters for Eastern Tibet grasslands, Sci. China Earth Sci., 55(6), 1001–1011, https://doi.org/10.1007/s11430-012-4433-0, 2012.

Yuan, L., Ma, Y., Chen, X., Wang, Y., Li, Z.: An enhanced MOD16 evapotranspiration model for the Tibetan Plateau during the unfrozen season, J. Geophys. Res. Atmos., 126, e2020JD032787, https://doi.org/10.1029/2020JD032787, 2021.

■ Dear Authors, I would like to bring to your attention that the data description provided is quite limited. Without a detailed description, including information such as the data format, it becomes challenging to thoroughly test and evaluate the data. The current description in the data repository seems more like an abstract. I kindly request that you provide a more thorough and detailed description of the data. Thank you for your attention to this matter.

Thank you very much for the suggestions provided by the editors. We gladly accept them. We have supplemented the data description as follows:

**Datasets Summary:**

This dataset provides gridded monthly evapotranspiration (ET) data for the Tibetan Plateau (TP) over the past 37 years (1981–2018), including soil evaporation ($E_s$), vegetation transpiration ($E_c$), and interception evaporation ($E_w$). The data have a horizontal resolution of 0.05° and are presented in mat format. To generate this dataset, the MOD16 evapotranspiration model (MOD16-STM) was optimized and developed using site-level flux observations

and soil texture information. Combining remote sensing data and reanalysis data, the MOD16-STM model was employed to simulate monthly evapotranspiration data for the TP region over the past 37 years. The results indicate that, compared to current mainstream gridded ET products in the TP region, this dataset exhibits higher accuracy. The dataset can be utilized for climate analysis, hydrological analysis, and relevant engineering applications.

**How to name and use data files:**

The data files are stored in separate folders based on the year. The data files are named YYYYMM_ET.mat, YYYYMM_Es.mat, YYYYMM_Ew.mat, and YYYYMM_Ew.mat, where YYYY represents the four-digit year and MM represents the two-digit month. The dataset is in Coordinated Universal Time (UTC). The unit is mm/month, indicating the total evapotranspiration for the respective period. The data can be read using Matlab. A simple example for reading the data is provided in the readme file for this dataset.

**Responses to Comments Made by Reviewer #1:**

The authors did a good job in revision. In particular, the dataset at 0.05 degree has now been produced rather than that at 0.01 degree. I have no further comments and suggest accepting it.

- A technical issue that needs to be corrected: In Line 271 and Table 3, the Shuanghu station needs to cite Ma et al. (2015b), already existing in the current Reference part.

Thank you very much for the reviewer's recognition of our research findings. In response to the questions you raised, I have already made the necessary revisions in the manuscript, which are highlighted in blue font. Please refer to Table 3 on line 279 for details.

**Responses to Comments Made by Reviewer #2:**

Thank you very much for the reviewer's recognition of our research findings.

**Responses to Comments Made by Reviewer #3:**

■ This is a resubmitted manuscript. The authors have addressed most of comments/suggestions and made changes to the paper, which has improved its quality considerably. Nonetheless, it seems that a few critical points were still ignored. A lot of empirical coefficients were cited from studies performed at different areas (e.g., equations 7 to 17), how large uncertainty did they cause, especially when estimating $r_s$ using $\theta_{sat}$ (even though the validation results seem good)? Insightful discussion on such problems with sensitive analyses may be good for understanding the possible accuracy of the products, including its components.

Thank you very much for the suggestions provided by the reviewer. We gladly accept them. In this paper, there are numerous empirical parameters in equations 7–17, all of which pertain to the parameterization of $r_a$ and $r_s$ in the MOD16-STM model. These parameters have already been assessed for their importance in ET estimation in the literature by Yuan et al. (2021). There are too many studies on investigating the empirical coefficients in equation 7–13. We will not repeat this analysis again. The uncertainty of $r_s$ to $a$ and $b$ in equation 14 has been reflected by Figure 3 in the manuscript. The parameterization method of $\theta_{sat}$ in the estimation of $r_s$ in this study is composed of various empirical parameters ($\rho_p$, $\rho_{soc}$, and $\theta_{sat, sc}$) for different soil types. We have conducted a uncertainty analysis of the estimated $\theta_{sat}$ and sensitivity of its uncertainty to the changes of empirical parameters. The impact of the empirical parameters on the estimation of ET is illustrated in Figure A1 (a–c). The results indicate that with a 20% uncertainty range in the estimated parameters $\rho_p$, $\rho_{soc}$, and $\theta_{sat, sc}$ for $\theta_{sat}$, the loss in estimating ET is only below 3%. The uncertainty of $\theta_{sat}$ to changes in these empirical parameters (equation 15–17) is relatively low. Thus, the conclusion is drawn that the estimation of ET is not sensitive to variations in these three parameters.

We have conducted a uncertainty analysis of the estimated $\theta_{sat}$. Figure A2 shows the accuracy of the estimated $\theta_{sat}$ by the method used in this paper. It demonstrates that the parameterized method of $\theta_{sat}$ could have a good representation of different soil types. Additionally, a sensitivity analysis is conducted on the empirical parameters $a$ and $b$ for calculating $r_s$. Keeping $\theta_{sat}$ and SM constant, $r_s$ exhibits exponential changes with variations in $a$ and $b$, leading to significant fluctuations in the estimation of ET. Within a 20% range of variation in $a$ and $b$, the maximum loss in ET exceeded 50% in Figure A1 (d–e). Therefore, it is essential to perform significance tests on the fitting results of the empirical parameters $a$ and $b$, as well as independent validation of the final ET estimates. The sensitivity test show that the factors that have a significant impact on $r_s$ and ET are the topsoil moisture and soil organic matter content. The following Figure A3 present the impact of soil organic matter content on $\theta_{sat}$ and ET estimation at different soil types. We also add discussions on these sensitivity analysis in the revised manuscript. Please refer to lines 207 to 230 and lines 647 to 675 in the manuscript for more details.

[Figure]

Figure A1. A sensitivity analysis on the impact of the uncertainty in the empirical parameters ($\rho_p$, $\rho_{soc}$, $\theta_{sat, sc}$, $a$ and $b$) of the estimated $\theta$sat on the estimation of ET (test in August, 2018).

[Figure]

Figure A2. Validation of the consistency between the estimated values and the observed values for $\theta_{sat}$ over the TP.

[Figure]

Figure A3. The sensitivity of LE and $\theta_{sat}$ to the changes of $m_{soc}$ content at different sites.

Chen, Y., Yang, K., Tang, W., Qin, J., and Zhao, L.: Parameterizing soil organic carbon's impacts on soil porosity and thermal parameters for Eastern Tibet grasslands, Sci. China Earth Sci., 55(6), 1001–1011, https://doi.org/10.1007/s11430-012-4433-0, 2012.

Yuan, L., Ma, Y., Chen, X., Wang, Y., Li, Z.: An enhanced MOD16 evapotranspiration model for the Tibetan Plateau during the unfrozen season, J. Geophys. Res. Atmos., 126, e2020JD032787, https://doi.org/10.1029/2020JD032787, 2021.

■    Moreover, I suggested that the authors compare they ET components with the other published datasets previously. Unfortunately, they said there's no publicly available data. As the author provide ET components datasets and results, I suggest the authors perform such comparison to reach the high level of the journal. Several observations (i.e., T/ET values) can be found in the following literatures (at least) (and/or using other remote sensing products such as GLEAM):

(1) Cui, et al. (2020). Quantifying the controls on evapotranspiration partitioning in the highest alpine meadow ecosystem. Water Resources Research, 56.

(2) Guo et al. (2017). River recharge sources and the partitioning of catchment evapotranspiration fluxes as revealed by stable isotope signals in a typical high-elevation arid catchment. Journal of Hydrology, 549, pp. 616-630.

(3) Zheng, C., Jia, L., Hu, G. (2022). Global land surface evapotranspiration monitoring by ET-Monitor

model driven by multi-source satellite earth observations. Journal of Hydrology, 613, 128444.

Thank you very much for the suggestions provided by the reviewer. We gladly accept them. First, in Section 4, we have divided it into two part, cross-validation of ET products and cross-validation of ET components. Please refer to lines 456 to 502 in the manuscript for more details.

**4.1 Cross-Comparison of the Spatial Distribution of ET on the TP**

[revised manuscript text omitted]

- Method section still contains inconsistency. First, $r_s$ and $r_c$ were corrected in Eq. 1. However, it seems they only use $r_s$ in the estimation without any explanation.

Thank you very much for the suggestions provided by the reviewer. We gladly accept them. Here, $r_a$ (s/m) is the aerodynamic resistance, $r_c$ (s/m) is the aerodynamic resistance of water vapor of the canopy, and $r_s$ (s/m) is the surface (or canopy) resistance. Yuan et al. (2021) optimized $r_a$ based on the Monin-Obukhov similarity theory (MOST) and calibrated the empirical values of $r_c$ for grassland underlying surfaces. They also pointed out that the topsoil moisture content directly affects the value of $r_s$, indirectly influencing the Es process. Therefore, this study extended this optimization algorithm from the site scale to the regional scale. Please refer to lines 141 to 146 in the manuscript for more details.

- Second, did the input datasets at 3h temporal scale include nighttime data (as well as the estimation)?

Thank you very much for the suggestions provided by the reviewer. The 3-hour interval data in the paper are considered as initial data, which includes nighttime data, and are subsequently averaged to daily or monthly scales.

- Thirdly, it is also unclear whether some variables are typo, such as $u_*$ in equation 13 is different from that in line 184.

Thank you very much for the suggestions provided by the reviewer. We gladly accept them. The variable "$u_*$" throughout the paper represents the friction velocity. We have made modifications in the paper. Please refer to Formula 13 and lines 184 to 186 in the manuscript for details.

- Finally, it is also unclear that where a few equations were from, such as those in lines 184 to 186?

Thank you very much for the suggestions provided by the reviewer. Formulas 8-12 are derived from the references in Högström (1996) and Paulson (1970). We have added citations of these papers for equation 8–12, to show their sources. Please refer to lines 178 to 184 in the manuscript for more details.

- Grammar and typos also need to be paid attention to across the entire manuscript, such as "-3-1 mm/year" in line 43.

Thank you very much for the suggestions provided by the reviewer. We gladly accept them. We have already corrected and modified the grammar and symbols in the manuscript. The modified portions are as follows: Our findings reveal a noteworthy upward trend in ET in most central and eastern parts of the TP, with

a rate of approximately 1–4 mm/year ($p<0.05$) and a significant downward trend with rates between −3 and 1 mm/year in the northwestern part of TP during the period from 1982 to 2018. Please refer to lines 41 to 44 in the manuscript for details.